

**Geosystemics and Earthquakes**
Angelo De Santis [1,2] , Gianfranco Cianchini[1], Rita Di Giovambattista[1], Cristoforo Abbattista[3],
Lucilla Alfonsi[1] , Leonardo Amoruso[3], Marianna Carbone[3], Claudio Cesaroni[1], Giorgiana De
Franceschi[1], Anna De Santis[1], Alessandro Ippolito[1], Dedalo Marchetti[1], Luca Martino[2],
Francisco Javier Pavòn-Carrasco[1,4], Loredana Perrone[1], Alessandro Piscini[1], Mario Luigi
Rainone [2], Luca Spogli[1,5] and Francesca Santoro[3]
[1] Istituto Nazionale di Geofisica e Vulcanologia – Sez. Roma 2, Rome (Italy)
[2] Università G. D'Annunzio, INGEO Deparment, Chieti (Italy)
[3] Planetek Italia srl, via Massaua 12, Bari (Italy)
[4] Now at Facultad Física (UCM), Avd. Complutense, s/n. 28040 – Madrid (Spain)
[5] SpacEarth Technology, Rome, Italy
**Abstract**
Geosystemics [De Santis 2009, 2014] studies the Earth system as a whole focusing on the possible
coupling among the Earth layers (the so called *geo-layers*), and using universal tools to integrate
different methods that can be applied to multi-parameter data, often taken on different platforms. Its
main objective is to understand the particular phenomenon of interest from a holistic point of view.
In this paper we will deal with earthquakes, considered as a long term chain of processes involving,
not only the interaction between different components of the Earth's interior, but also the coupling
of the solid earth with the above neutral and ionized atmosphere, and finally culminating with the
main rupture along the fault of concern [De Santis et al., 2015a]. Some case studies (particular
emphasis is given to recent central Italy earthquakes) will be discussed in the frame of the
geosystemic approach for better understanding the physics of the underlying complex dynamical
system.



## 1. Introduction

The present times are difficult for humankind: although civilization, together with technology, has
reached its greatest level, the cost to pay is that modernization has simultaneously involved some
level of higher vulnerability than in the past (e.g. Nott, 2006). Humankind is affecting some parts of
Nature, but, unfortunately, without perfect control. Natural hazards, e.g. hurricanes, earthquakes
(EQs), floods, tsunamis, and other kinds of catastrophes, are often out of human control and the
consequences are unpredictable. They  happen as extreme events on the planet causing destruction
and deaths (Meyers, 2009), and the occurrence of most of them looks as increasing dramatically in
the last century (Peduzzi, 2005). Present society, although reaching a high technological level, is
still very weak, and no strong remedy and rapid resilience are fully possible (Bunde et al., 2002).
There is no other solution than attempting to understand how our planet works and what possible
future sceneries are. To do this, we cannot limit our approach to a reductionist one, but we also need
to study the Earth as a whole system, where all parts are nonlinearly interconnected and useful for
the system to evolve (e.g. Skinner and Porter, 1995). In contrast with the reductionist approach, the
latter does not look at Earth as a precise clock system where all components have their distinct own
purpose (often called as the Laplacian point of view), rather we have to consider it as an ensemble
of cross-interacting parts put together in order to reach the same ambitious goal that, at the present
knowledge, seems to be rare in the Universe: to maintain life (Lovelock, 1972). Earth system is
both composed of living organisms and soft and hard engines, in a continuous balance and
competition between life and death, heat and cold, complexity and simplicity, chaos and non-chaos.
In this paper we will remind the concepts of Geosystemics and then apply it to EQs, through some
specific concepts such as the changes of Benioff strain, Entropy, temperature, etc., in the frame of a
Lithosphere-Atmosphere-Ionosphere (LAI) coupling model, i.e. some quantities that are related to
macroscopic features of the system under study.
Even if many efforts have been made towards a deeper knowledge of EQs, in terms of experimental,
theoretical and numerical models (e.g. Aki & Richard, 2002; Bizzarri, 2011, 2012), their evolution
phases are not exhaustively explained yet. A possible explanation of this uncertainty is the lack of
knowledge regarding the source initiation, the fracture mechanisms and dynamics of the crust (e.g.
Scholz, 2002). Moreover, each EQ initiates and develops in its proper geodynamical and



lithological settings, this giving an almost unique character to each event. Thus, to reach the
knowledge necessary to recognise in advance the eventual rupture (failure) of the fault which causes
the occurrence of the EQ is a greatly difficult task itself. Difficult as well is the possible explanation
of the various and often weak phenomena affecting the above atmosphere and ionosphere, where
even many external causes act to mix together signals which are different in spectral content and
amplitudes.
Despite all these difficulties, one of the most eminent seismologists, Hiroo Kanamori (1981),
pointed out that some common physical mechanisms beneath the generation processes may act,
although controlled by the local geodynamic forces and heterogeneities of the lithology (Baskoutas
and Papadopoulos, 2014): this thought encourages the efforts towards a deeper knowledge of the
physics behind such a complex phenomenon as the EQ.
If the process of rupture that causes the EQ is still plenty of open issues and unanswered questions
(e.g. Bizzarri, 2014), even more difficult is the understanding of the process of EQ preparation,
although some efforts have been performed (e.g. Dobrovolsky et al. 1979, 1989; Sobolev et al.
2002) as is thought to be accompanied by some exchanges of mass and energy, which can change
the energy budget in the earth-atmosphere system over the seismogenic zone and, in fact, scientific
literature reports a wide variety of phenomena preceding EQs which have been studied extensively
with the aim of finding some recurrent and recognizable patterns: induced electric and magnetic
fields, groundwater level changes, gas and infrared (IR) electromagnetic emissions, local
temperature changes, surface deformations (see Cicerone et al., 2009 and De Santis et al., 2015a for
more exhaustive reviews).
Being Geosystemics introduced by one of the present coauthors, great part of this paper is based on
own contributions from mostly already published material. However we attempted to give new
insights on the idea of Geosystemics with also some unpublished own material or other researchers'
contributions.
At first, we place the present view of Geosystemics and show the application to some case studies.
We also present a possible physical model that attempts to explain the found results. We then
conclude with some feasible future directions and conclusions.



**2. Geosystemics**
We depict here the general concepts of geosystemics in order to be then applied to EQs in the next
section.
Geosystemics looks at the Earth system in its whole: in particular, it focuses on self-regulation
phenomena and relations among the parts composing the Earth. Dynamically speaking, it also
searches for the possible trends of change or persistence of the specific system or sub-system under
study (De Santis, 2009, 2014, De Santis and Qamili, 2015). To perform this, geosystemics applies
mainly the concepts of entropy and information content to time series characterising the
phenomenon under study: to measure and understand the physical world, not only energy and
matter are important, but also information (Bekenstein, 2003). Interesting features of the complex
system under study to investigate are: nonlinear coupling and new emergent behaviour,  self-
regulation, and irreversibility as important constituents of the Earth planet.
The most important basic concept of geosystemics is that no layer of the Earth system is really
isolate, rather it communicates (in terms of transfer of energy or particles) with the other ones. This
concept is more strengthen in case of very powerful phenomena that release a large energy in a
short time, such as the earthquakes in the lithosphere (for a M7 earthquake, around $10^{15}$ Joule are
released in some seconds), the lightning strikes in atmosphere (around $10^9$ J in microseconds),
etcetera. For instance, the information exchanged between contiguous parts of the Earth system
producing increased entropy would allow us to better recognise and understand those irreversible
processes occurring in the Earth's interior. As said in De Santis and Qamili (2015), "*Geosystemics*
*has the objective to observe, study, represent and interpret those aspects of geophysics that*
*determine the structural characteristics and dynamics of our planet and the complex interactions of*
*the elements that compose it*" by means of some entropic measures.
Together with this, the approach will be based on multi-scale/parameter/platform observations in
order to better scrutinise the particular sub-systems of Earth under study as much as possible. This
is a fundamental issue of geosystemics, because there is no better way to understand the behaviour
of a complex system than looking at it from as many perspectives and points of view as possible.
Recent advanced examples to observe the planet are from satellites (e.g. Chuvieco and Huete, 2009)
and seafloors (Favali et al., 2015).



Geosystemics differs from the standard Earth System Science (e.g. Schneider and Boston, 1992;
Skinner and Porter 1995; Jacobson et al., 2000; Butz, 2004): for instance, in the way it is applied by
means of entropic measures to different physical quantities, this because entropy is the only entity
that can be used to have some clues on the next future (please remind the second law of
thermodynamics; e.g. Grandy, 2008).
In this paper, we will concentrate the attention to the application to EQ physics study and the
possibility for intermediate and/or short-term prediction. Here, with the term "prediction", we mean
the possibility to make a prediction about EQ occurrence, magnitude and location, with small
uncertainty, i.e. in a deterministic way, in contrast with the probabilistic approach used in EQ
forecast (please see also in the next section for other details on this question). In particular, we will
explore the present state-of-the-art of the seismological diagnostic tools based on a macroscopic
point of view. Particular emphasis will be dedicated to the above mentioned Shannon entropy. Later
on, we also see another one that quantifies the sense of flow of information, the transfer entropy
(Schreiber, 2000).
**3. Main seismological diagnostic tools**
The Holy Grail in seismology is to reach the capability of giving short-term prediction of large EQs
thus eventually saving lives. Unfortunately, it is not an easy task as testified by the great all-out and
full scale effort made with this aim in many fields of research (even far from the traditional field of
seismology) and the corresponding huge amount of scientific papers claiming or denying success or
simply attempting some important steps forward towards the goal. However, despite of the many
attempts no significant success has been clearly counted (Hough, 2009).
Regarding the methods to make EQ "predictions", we can classify them in (mainly) *deterministic*
and (mainly) *stochastic* methods. The bracket term "(mainly)" is placed because, actually, no
method is only deterministic or stochastic. To be operative, we can define the latter methods as
those that provide a forecast with some level of probability, for which the probability of no EQ is
always different than zero, while the deterministic methods attempt to indicate the approaching of a





large EQ with some level of confidence, i.e. with small uncertainty in space and time of occurrence,
and magnitude.
Several statistical methods have been applied in the last decades to seismological data (mainly
catalogs) with the aim of improving the knowledge on seismic phenomena. At present, the scientific
community is involved in global projects to test and evaluate the performances of some well
established algorithms in different tectonic environments (see http://www.cseptesting.org/;
http://www.corssa.org/ websites). According to CSEP (Collaboratory for the Study of Earthquake
Predictability), the most important steps of an earthquake prediction protocol are the following
ones:
1. *Present a physical model that can explain the proposed precursor anomaly.*
2. *Exactly define the anomaly and describe how it can be observed.*
3. *Explain how a precursory information can be translated into a forecast and specify such a*
*forecast in terms of probabilities for given space/time/magnitude windows.*
4. *Perform a test over some time that allows to evaluate the proposed precursor and its forecasting*
*power.*
5. *Report on successful prediction, missed earthquakes, and false predictions.*
In this part, we will focus our attention to the deterministic methods, which are essentially based on
a systematic catalog-based search of some peculiar *seismicity pattern recognition* in the given area
of interest. A wide review on this topic is presented by Mignan (2008). In the following, we will
describe *M8*, *RTP* (Reverse Tracing of Precursors), *PI* (Pattern Informatics) and *R-AMR* (Revised
Accelerating Moment Release). The latter method is the most recent and is the one we know much
better, because some of the present co-authors have introduced the corresponding technique (De
Santis et al., 2015b). For this reason, we will dedicate a specific section to it.
*M8*
*M*8 was called in this way because it was designed by retroactive analysis of the seismicity
preceding the greatest (M8+) EQs worldwide (e.g. Keilis-Borok and Kossobokov, 1990;
Kossobokov, 2013). Some spatio-temporal functions are introduced in order to reconstruct a 7-

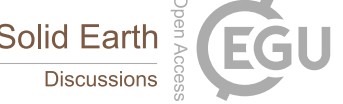



dimensional phase space (the first three quantities are estimated for two values $C$ of the average
annual number of earthquakes in the sequence; usually C=10 and 20): $N(t)$ is the number of main-
shocks; $L(t)$ the deviation of $N(t)$ from the long-term trend; $Z(t)$ the linear concentration of the main-
shocks; and $B(t)$ is the maximal number of aftershocks as a measure of EQ clustering (these
functions are actually properly normalized). The algorithm then recognizes a well established
criterion, defined by extreme values of the phase space coordinates, as a vicinity of the system
singularity. When a trajectory enters the criterion, probability of extreme event increases to the level
sufficient for its effective provision, so an alarm or a TIP, "Time of Increased Probability", is
declared. This algorithm can be adapted for lower magnitudes and particular regions (e.g. CN8).
*The Reverse Tracing of Precursors (RTP)*
The *RTP* is a method for medium-term (some months in advance) EQ prediction (Shebalin et al.,
2006), which is based on a hierarchical ensemble of premonitory seismicity patterns. These patterns
are: (1) "precursory chains" that are related with the correlation length (e.g. Zoller and Hainzl, 2002,
Tyupkin and Di Giovambattista, 2005), (2) "intermediate-term patterns" that could be related to
some accelerating seismicity (e.g. Gabrielov et al., 2000) and (3) "pattern recognition of infrequent
events" that take into account several "opinions" to decide the validity of the calculated chain of
events. If a sufficient number of "votes" is accumulated, then the chain is considered precursory
(Shebalin et al., 2006). Some past EQs seem to have been predicted 6 to 7 months in advance,
although a few false alarms also happened. Critical aspects are related to the predicted "area of
alarm" that seems very large for a realistic application.
*Pattern Informatics (PI)*
The *PI* is a technique for quantifying the spatio-temporal seismicity rate changes in historic
seismicity (e.g. Rundle et al., 2002; Nanjo et al. 2005). Tiampo et al. (2006) derive a relationship
between the "PI index" and stress change (e.g. Dieterich, 1994), based upon the crack propagation
theory. In practice, the PI method measures the change in seismicity rate at each box of a pre-
defined grid, relative to the background seismicity rate, through the division of the average rate by
the spatial variance over all boxes. Then it identifies the characteristic patterns associated with the
shifting of small EQs from one location to another through time prior to the occurrence of large
EQs (Tiampo et al., 2006). Results are given in terms of mapping the "PI anomalies" which are



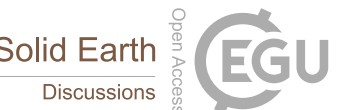

located where a new large EQ can be expected. Holliday et al. (2007) have proposed a modification
of PI by using complex eigenfactors, explaining the EQ stress field as obeying a wave-like equation.
**4. Shannon Entropy and Shannon Information**
The Shannon Entropy $h$(t) (Shannon, 1948) is an important tool for the space-time characterization
of a dynamical system. In general, for a system characterized by $K$ possible independent states, this
entropy is defined in a certain time $t$ as follows:

$$h(t) = -\sum_{i=1}^{K} p_i(t) \cdot \log p_i(t) \qquad (1)$$

where $p_i(t)$ is the probability of the system to be at the $i$-th state. For convenience, we impose
$\sum_i p_i = 1$ and $\log p_i = 0$ if $p_i = 0$ to remove the corresponding singularity.
In literature, we can find a large number of physical interpretations of the Shannon Entropy. We
consider here what we think it is the simplest one: it is a non-negative measure of our ignorance
about the state of the system of concern. The Shannon Entropy has a great importance in evaluating
and interpreting the behaviour of complex systems like Earth, in general, and EQs, in particular. On
the other hand, we find in literature also the Shannon Information, $I(t)$, which is simply related to
$h$(t) as $I$(t)=- $h$(t). Consequently, the Shannon information is a negative quantity that measures our
knowledge on the state of the system when we know only the distribution of probability $p(t)$ (Beck
and Schlögl, 1993). Thus, this quantity measures our decreasing ability to predict the future
evolution of the system under study.
**5. Gutenberg-Richter law and $b$-value**
The Gutenberg-Richter (GR) law has a central role in seismology (Gutenberg and Richter, 1944). It
expresses the (decimal logarithm of the) cumulative number $n$ of EQs with magnitude $m$ equal to or
larger than a magnitude $M$:

$$\log n(m \geq M) = a - bM \qquad (2)$$





as a simple linear function of the magnitude $M$; $a$ and $b$ are two constant parameters for a certain
region and time interval, characterizing the associated seismicity; in particular, $b$ is the negative
slope of the above cumulative distribution and typically $b \approx 1$. Very soon it was recognized the
importance of estimating the $b$-value as an indicator of the level of stress in a rock from laboratory
experiments (Scholz, 1968), and only later the relationship has been confirmed for EQs
(Schorlemmer et al., 2005).
Aki (1965) was the first to provide a simple expression to estimate $b$ by means of the maximum
likelihood criterion (with a correction proposed by Utsu, 1978):

$$b = \frac{\log e}{\overline{M} - M_{min} + \Delta/2}$$

9                                                                       (3)

with an uncertainty of $\pm b/\sqrt{N}$; $N$ is the total number of analyzed EQs; $e = 2,71828 \ldots$ is the Euler
number, while $\overline{M}$ is the mean value of the magnitudes of all considered EQs; $M_{min}$ is the minimum
magnitude used in the $b$-value evaluation; $\Delta$ is the resolution involved in the magnitude estimation,
normally $\Delta=0.1$. Usually the $M_{min}$ is the magnitude of completeness of a seismic catalog, i.e. the
magnitude threshold at which or above the corresponding seismic catalog includes all occurred EQs
in the region.

**17   6. Entropy and EQs**

Now we apply the concept of Shannon Entropy to EQs. Most of this section is based on De Santis
et al. (2011a) with some extension in order to clarify some concepts.
Given a sequence of EQs (in the form of a seismic catalog or a seismic sequence within a certain
region) with non-negative (and normalized) probability $P_i$ to have activated a certain $i$-th class of
seismicity characterized by some range of magnitudes, the associated non-negative Shannon
entropy $h$ can be defined as (De Santis et al., 2011a):

$$h = -\sum_{i=1}^{K} P_i \cdot \log P_i \geq 0$$

27                                                                      (4)

Since eq. (4) is applied to a discrete number of states, $h$ is also called *discrete Shannon entropy*. It
can be considered a reliable measure of uncertainty and missing information of the system under
study.




Actually, the values of magnitude can assume a continuous range (in theory from $M_{min}$ to infinity)
then the discrete definition (4) becomes an integral definition (Shannon, 1948; Ihara, 1993):

$$H = - \int_{M\min}^{\infty} p(M) \cdot \log p(M) dM$$

3                                                                                                      (4a)

where $H$ is now called *continuous* (or *differential*) *Shannon entropy* to be distinguished from $h$, and
$p(M)$ is the probability density function (*pdf*) of the magnitudes $M$, such as

$$p(M) = \frac{d}{dm} \sum_{i,m \leq M} P_i(m)$$

6                                                                          ,

and

$$\int_{M\min}^{\infty} p(M) dM = 1$$

It is worth noticing that moving from the discrete definition (4) to the continuous (4a), the property
of non-negative $h$ is lost by $H$, that can assume also negative values (e.g., Ihara, 1993). This is not
evident from the work by De Santis et al. (2011a), so we will spend here some words about it.
The two definitions of the Shannon entropy are related by the following equation:

$$H = h + \log \Delta$$

15                                                                                                      (5)

where $\Delta$ is the sampling step of the continuous *pdf* in order to let it discrete in $h$ (e.g. Klir 2006). It
is evident from (5) that when $\Delta$ tends to zero, $H$ will diverge to $-\infty$. Thus, the continuous entropy $H$
is not a limit for $\Delta \to 0$ of the Shannon discrete entropy $h$ and, consequently, it is not a measure of
uncertainty and information. However, the continuous Shannon entropy can be used to measure
differences in information (Ihara 1993).
However, when the classes of magnitude are loose, e.g. for $\Delta \approx 1$ we will have $H \approx h$: De Santis et al.
(2011a) considered $\Delta = 0.5$ so the difference between discrete and continuous entropies was only 0.3.
De Santis et al. (2011a) have shown that if the $p(M)$ is the GR probability distribution, then $H$ can
be expressed in terms of the *b*-value:

28                              $H \approx 0.072 - \log b$                                            (6)

This relation is detailed and proved in the Appendix.





**7. Entropy and critical point theory**
An *ergodic* dissipative system can have a critical point where the system undergoes through a
transition. The ergodic property means that the system averages in real 3D space are equivalent to
averages in the ideal reconstructed phase space (e.g. Takens, 1981, De Santis et al., 2011b). As an
example, Fig. 1 reports the behaviour of the *specific heat* around a critical point occurring at
temperature $T_\lambda$ degree. It is interesting how the system approaches the critical temperature as a
power law. In addition, if the system changes its temperature linearly in time, the same plot is
expected versus time.

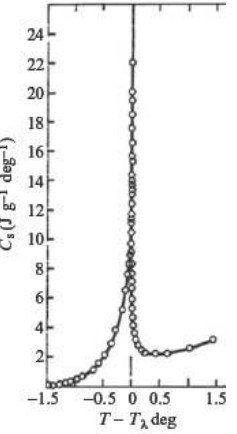

**Fig. 1**. Specific heat of $^4$He as a function of $T$-$T_\lambda$ in Kelvin. $T_\lambda$ is the temperature at which the critical system has a
transition (critical point) (Adapted from Stanley, 1971).
More generally, if we replace the increasing temperature with the system entropy, then the system
reaches its critical point (vertical red line in Fig.2) at the largest Entropy and approaches it with an
accelerating power law in its cumulative of *punctuated* events (we intend here for an "event" as an
anomalous behaviour of the system evolution, e.g. when its signal level is larger than a certain
number of standard deviation, $\sigma$, e.g. 2.5 $\sigma$).





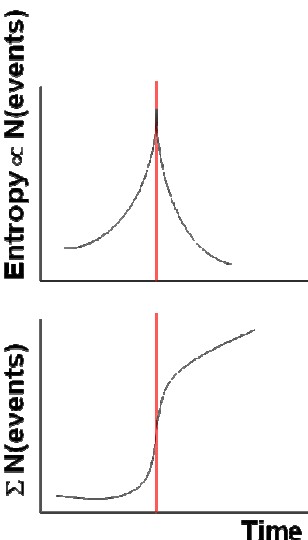

**Fig. 2**. Idealized Shannon entropy (above diagram) and cumulative number of events (bottom diagram) for a dissipative
system around its critical point, indicated by the vertical red line.
After the critical point, the curve behaves as a decelerating power law. Fig. 2 depicts both the
idealized behaviours for the entropy and the cumulative number of events.
We will see in the following how these patterns are reproduced in the case studies of some Italian
seismic sequences.
**8. Entropy studies of two Italian seismic sequences**
In this part, we show two case studies in Italy: the 2009 L'Aquila and the 2012 Emilia seismic
sequences, both producing a main-shock of around M6 (precisely local and moment magnitudes,
ML5.9 and Mw6.2 for L'Aquila and local magnitude ML5.9 for Emilia). Main characteristics of the
two seismic sequences are given in Table 1. The first case was already analysed and discussed by
De Santis et al. (2011a). However, we will make here some alternative/complementary analyses,
with respect to those already published. The second case study is original and never published so far.



| Sequence ID | Main-shock Parameters | | | | | | # data (foreshocks) | $R_{max}$ (km) | $M_{min}$ |
|---|---|---|---|---|---|---|---|---|---|
| | Coord. (lat, lon) in degree | Depth (km) | Date | $t_f$ in days from 1 May 2005 (predicted) | Fault style | Magnitude (predicted) * | | | |
| L'Aquila | 42.34N 13.38E | 8.3 | 6 Apr 2009 | 1436.06 (1437.4) | N | 5.9 (5.3±0.5) | 17 | 300 | 4.0 |
| Emilia | 44.89N 11.23E | 6.3 | 20 May 2012 | 2576.09 (2577.7) | R | 5.9 (5.7±0.5) | 38 | 300 | 4.0 |

* normally deduced from eq. (2a or 2b)

**Table 1** - Main data related to the two Italian seismic sequences under study: (from left to right) the label, the main-
shock source parameters, the number of data points (foreshocks) used in the fitting stage; the maximum distance from
the main-shock epicentre defining the selection area and the minimum threshold magnitude of the selected events there
considered. We provide also a rough estimation of the predicted magnitude (within brackets) of the impending main-
shock (see text). N and R in the Fault style column stand for Normal, and Reverse focal mechanism, respectively. Rmax
and Mmin are the largest area and minimum magnitude, respectively, considered in the analyses.
*The 2009 L'Aquila seismic sequence*
As mentioned in De Santis et al. (2011a), Shannon entropy can be estimated in three different ways:
cumulative, moving overlapping or distinctively temporal windows. For the first case study, i.e. the
2009 L'Aquila (Central Italy) seismic sequence, we will consider adjacent non-overlapping moving
windows. In Fig.3 we show the estimation of the Shannon Entropy based on non-overlapping
windows of 30 M1.4+ seismic events occurred in a circular area of 80 km around the main-shock
epicentre. The low number of events used for the analysis in each window was chosen to better
follow even shorter fluctuations of entropy, especially for the foreshocks. It is interesting that two
distinct entropy values before the main-shock occurrence are larger than the threshold $H_t=2.5$ σ (the
mean value of entropy, $\langle H \rangle$, is practically zero). To better visualize the mean behaviour of entropy,
the gray curve defines a reasonable smoothing of the entropy values: 15-point FFT before the main-





shock and 50-point FFT smoothing after the main-shock. The different kind of smoothing is related
to the different rate of seismicity before and after the main-shock.  It is interesting to notice that the
smoothed gray curve of the Shannon Entropy reproduces the expected behavior of a critical system
around its critical point (as shown in Fig.1, but with time as *x*-axis ), with the main-shock as critical
point.

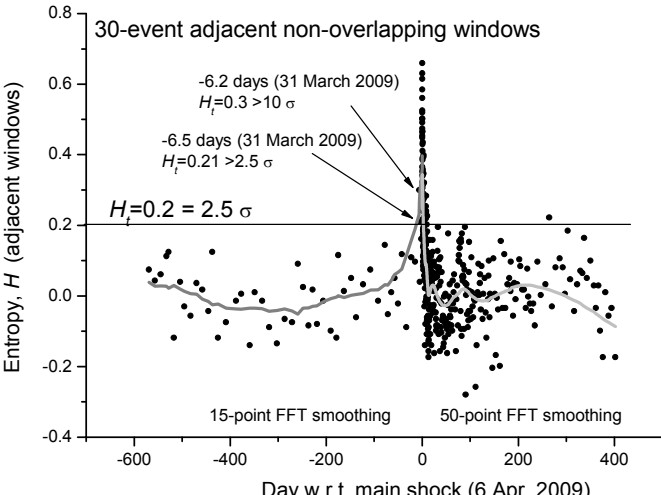

**Fig. 3**. Shannon entropy for L'Aquila seismic sequence from around 1.5 year before the main-shock to around 1 year
after, calculated for a circular area of 80 km around the main-shock epicenter. Each point is the entropy analysis based
on non-overlapping windows, each composed by 30 foreshocks. The gray curve defines a reasonable smoothing of the
entropy values: 15-point FFT before the main-shock and 50-point FFT smoothing after the main-shock. The different
kind of smoothing is related to the different rate of seismicity before and after the main-shock. It is interesting how the
smoothed curve reproduces the expected behavior of a critical system around its critical point. (Adapted from Wu et al.

14   2016).

We can even analyse in more detail the same curve, but expanded in the period before the main-
shock (Fig. 4). We confirm that, around 6 days before the main-shock, there is the persistence of
two consecutive values of entropy greater than $2.5\sigma$  (the larger value is  even greater than $10\sigma$). An
interesting question to better investigate in more case studies will be: could this persistence of larger
values of entropy be considered a reliable precursor of the imminent main-shock?





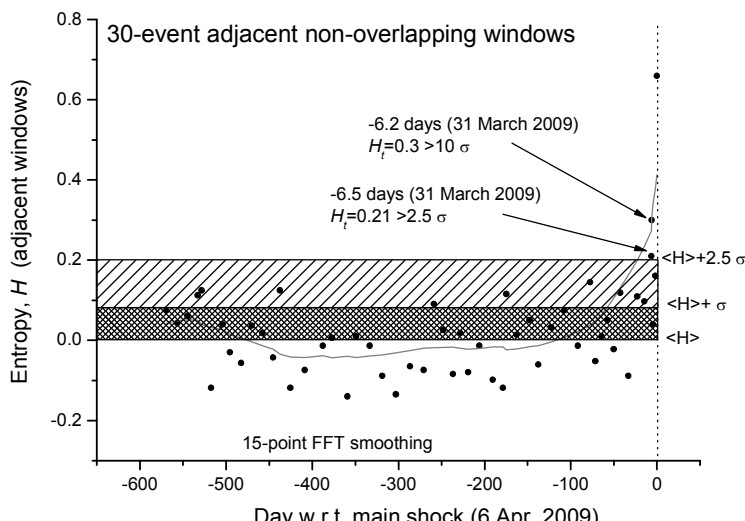

**Fig. 4**. Detail of the Shannon entropy for L'Aquila seismic sequence from around 1.5 year before the main-shock to the main-shock occurrence. Each point is the entropy analysis based of non-overlapping windows, each composed by 30 foreshocks. The mean value of the entropy, <H>, which is almost zero, and one and two standard deviations are also shown. The gray curve defines a reasonable smoothing of the entropy values with 15-point FFT.

*The 2012 Emilia seismic sequence*

In this specific case study, we will consider moving and partially overlapping windows, each composed of around 200 seismic events and overlapping of 20 events. This kind of analysis allows us to have directly a smoother curve of entropy, without resorting to a subsequent smoothing operation as done instead in the previous case.

In Fig. 5 we plot the Shannon entropy for Emilia seismic sequence from 2000 to 2014, as estimated over all M2+ EQs occurred around 150 km the first major EQ. The significant increase starting around 2010 is probably real and related to the preparation phase of the two major EQs occurred on 20 and 29 May, 2012 with local magnitudes 5.9 and 5.8, respectively, where the entropy reaches the maximum value (in this case around 0.3). The gray area defines the estimated error in computing the entropy.

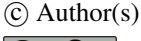
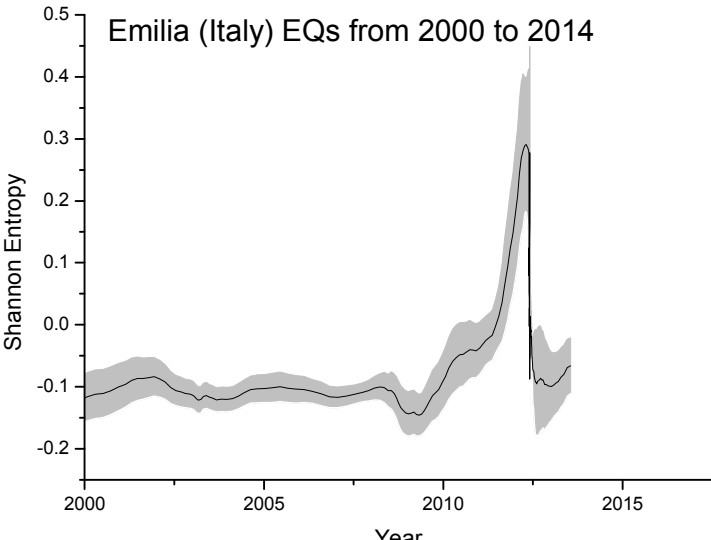

**Fig. 5**. Shannon entropy for Emilia seismic sequence from 2000 to 2014. The significant increase from around 2010,
with the maximum at around the main-shock occurrence, is expected to be real. The gray area defines the statistically
estimated (1 standard deviation) error in computing the entropy.
**9. Accelerated Moment Release revisited: the case of L'Aquila and Emilia EQs**
Benioff (1949) proposed a simple way to estimate the strain-rebound increment, $\varepsilon_i$ :

$$s_i = \sqrt{E_i} = k_i \cdot \varepsilon_i \tag{7}$$

where $E_i$ is the energy released by the EQ, i.e. $10^{\alpha M+\beta}$ ($\alpha$=1.5, $\beta$=4.8 for energy expressed in Joule,
although Benioff used slightly different values), and $k_i=(\mu P V_i/2)^{0.5}$ ($\mu$=shear or rigidity modulus,
$V_i$= volume of the i-th fault rocks, $P$ is the fraction of energy transmitted in terms of seismic waves;
usually it is considered $P \approx 1$). This theory is based on Reid (1910) arguments of the elastic rebound.
To take account of the cumulative effect of a series of $N$ EQs at the time $t$ of the last $N$-th EQ,
Benioff introduced therefore what is now called *cumulative Benioff strain s*:

$$s(t) = \sum_{i=1}^{N(t)} s_i = \sum_{i=1}^{N(t)} \sqrt{E_i} = 10^{\beta'} \sum_{i=1}^{N(t)} 10^{0.75 M_i} \tag{8}$$

with $\beta'= \beta/2= 2.4$. It is important to notice that, according to Benioff (1949), the cumulative strain
(8) is that accumulated *on the fault* under study.



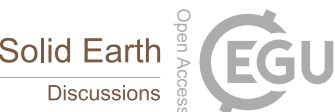

Extending the meaning of (8) to the strain accumulated over a larger area around the epicentre, Bufe
and Varnes (1993) obtained interesting results with the so-called *Accelerating Moment Release*
(*AMR*) approach that consists in fitting the cumulative value *s(t)* expressed as in (8), with a power
law in the time to failure $t_f$, i.e. the theoretic time of occurrence of the main shock: $s(t)=A+B(t_f - t)^m$,
where *A*, *B* and *m* are appropriate empirical constants (*m* is expected between 0 and 1: typical value
is 0.3; Mignan, 2008). The fitting process gives as an outcome the time $t_f$ together with the
expected magnitude, which is related to either *A* or *B*:

$$M_p(A) = \frac{\log(\Delta s_{last}) - \beta'}{0.75}$$
(9a)

where $\Delta s_{last} = A - s_{last}$ and $s_{last}$ is the cumulative Benioff strain at the last precursory event
considered (namely the *N*-th EQ). In this expression one speculates that the main-shock will be the
next EQ striking after the *N*-th, but the occurrence of many smaller EQs after the last analysed
shock and before the predicted time $t_f$ cannot be excluded.
An alternative formulation, based on the parameter *B,* has been given by Brehm & Braile (1999):

$$M_p(B) = \frac{\log|B| - \beta' - 0.14}{0.738}$$
(9b)

A criticism to this method came from Hardebeck et al. (2008) who pointed out the arbitrariness in
the critical choice of the temporal and spatial criteria for data selection, i.e. the initial precursory
event of the *AMR* curve and the extension of the inspected region.
To circumvent this criticism, De Santis et al. (2015) introduced what they called R-AMR, i.e. the
Revised Accelerating Moment Release (R-AMR), as a better way of applying the AMR by
weighting the EQs magnitudes in a certain area, according to an appropriate attenuation function
$G=G(R)$, where *R* is the distance of a given EQ epicentre with respect to the *impending slipping*
*fault*. In particular, the Benioff Strain produced at the fault level is expressed by a reduced Benioff
strain $\hat{s}(t)=s \cdot G$ called "reduced" because the action of the function *G*, which is normally less than
unity (i.e. $G \leq 1$), is to lower the value of the typical Benioff strain, normally according to the
distance *R* from the centre of the region of study. As area of interest, a circle is taken with the
corresponding Dobrovolsky radius, $r(km)=10^{0.43M}$ with M=EQ magnitude (Dobrovolsky et al.,

29  1979).

Thus, the expression for the cumulative *reduced* strain becomes:





$$\tilde{s}(t) = \sum_{i=1}^{N(t)} \tilde{s}_i = \sum_{i=1}^{N(t_i)} \sqrt{E_i} \cdot G(R_i) = 10^{\beta'} \sum_{i=1}^{N(t_i)} 10^{0.75 M_i} G(R_i) \qquad (8b)$$

De Santis et al. (2015b) applied with success their revisited method to the three most important
seismic sequences occurred in Italy in the last ten years. In addition, they also showed that, for a
particular seismic swarm (i.e. with no mainshock), R-AMR performs better than AMR, not
providing a false alarm.
*R-AMR for the 2009 L'Aquila and 2012 Emilia seismic sequences*
We show here two case studies, L'Aquila and Emilia seismic sequences, where the application of
R-AMR is made much simpler than the one firstly proposed in De Santis et al. (2015b). Although
that way of applying of R-AMR is more rigorous because all EQs above the minimum magnitude of
completeness are considered, we show here that a simpler application is possible, where considering
a very simple attenuation function of the form $G(R_i) = d/R_i^\gamma$, with d (normally 1km), $R_i$ in km and
with $\gamma \approx 1$, at the cost of considering a larger minimum magnitude threshold of around M4. Figs 6a
and 6b show the results for the cases of L'Aquila and Emilia sequences, where we apply to all
shallow (depth h≤40 and h≤80 km, respectively) M4+ EQs both AMR and R-AMR analyses (top
and bottom of each figure, respectively). Then, we then consider a 300 km size for the regions
where we applied R-AMR analysis. This size is comparable with the corresponding Dobrovolsky's
radius. Both the analyses stop well before the main-shocks that are not so considered in the
calculations. We notice that the time of preparation is rather long for both sequences, i.e. practically
starting at the beginning of the whole period of investigation (May 2005). This fact could be simply
interpreted as the larger foreshocks anticipate the beginning of the seismic acceleration with respect
to the smaller ones, which were the most in the previous analyses in De Santis et al. (2015b).
The goodness of the power law fit with respect to the linear regression can be quantified by the C-
factor which is the square root of the ratio between the RMS of the power law and the RMS of the
best linear fit (Bowman et al., 1998): the lower the C-factor than 1, the better the power law fit is
with respect to the line.





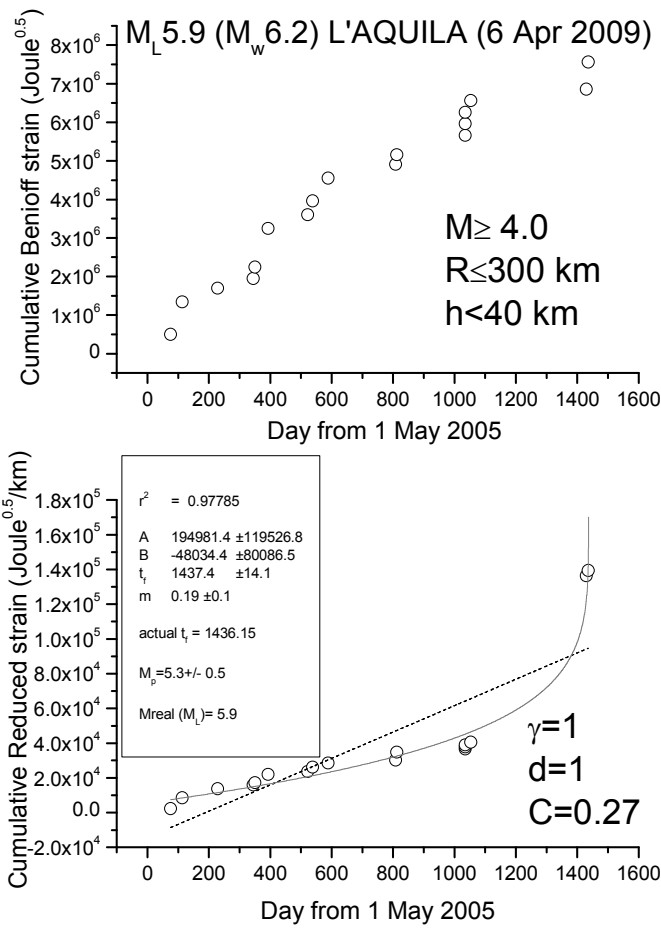

**Figure 6a —** Analyses of L'Aquila seismic sequence $M{\geq}4$ EQs (main-shock not shown and not used in the analysis):
top) ordinary AMR method; bottom) R-AMR method. Dashed line represents the best linear fit, while solid gray curve
is the best power law fit. Results of the fit are shown in the frame inside the graph at the bottom; $r^2$ is the coefficient of
determination, providing a measure of the quality of the fit (the closer to 1, the better the fit).
We find a clear seismic acceleration for both seismic sequences, quantified by a low value of C
(0.27 for L'Aquila sequence and 0.46 for Emilia sequence) and a great determination coefficient
($r^2>0.95$ in both cases). In addition, the predicted magnitudes are comparable with (although lower
than) the real ones. In both cases, the beginning of clear acceleration starts around 1.5 year before
the main-shock.



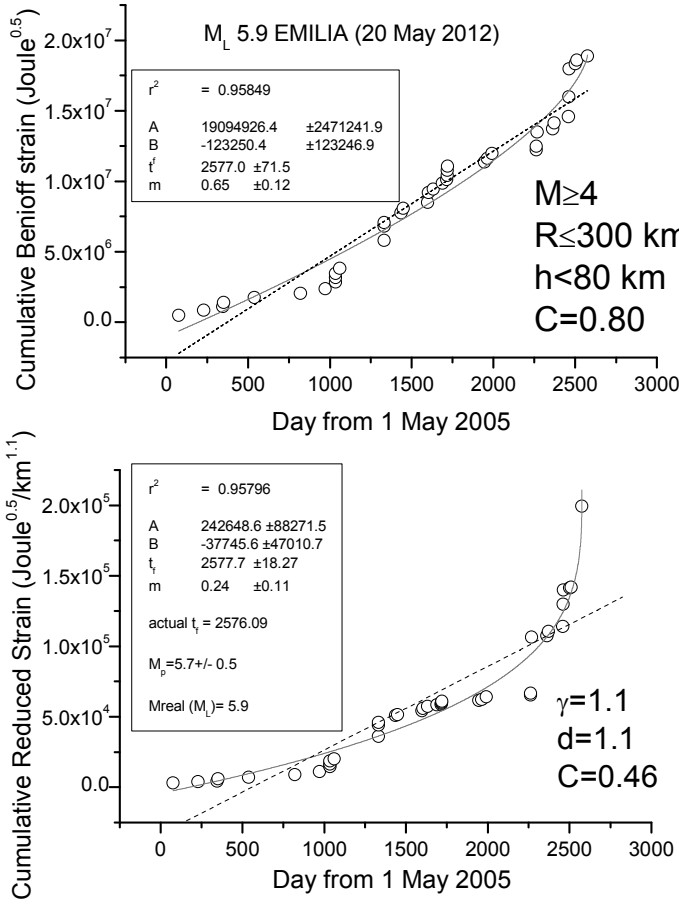

**Figure 6b —** Analyses of Emilia seismic sequence $M{\geq}4$ EQs (main-shock not shown and not used in the analysis): top)
ordinary AMR method; bottom) R-AMR method. Here the ordinary ASR also showed a little acceleration (*C*-
factor=0.80) but the R-AMR version is much better (*C*-factor=0.46). Dashed line represents the best linear fit, while
solid gray curve is the best power law fit. Results of the fit are shown in the frames inside the graphs; $r^2$ is the
coefficient of determination, providing a measure of the quality of the fit (the closer to 1, the better the fit).
**10. Lithosphere-Atmosphere-Ionosphere Coupling (LAIC)**
Geosystemics sees the planet in its entirety, where all geo-layers "communicate" each other, in
terms of exchange of matter and/or energy, i.e. what Bekenstein (2003) called with the more generic
term of "information". An important recent model is the so-called Lithosphere-Atmosphere-





Ionosphere Coupling (LAIC) for which before a large EQ, in its preparation phase, some precursory
anomalies can appear in atmosphere and/or in ionosphere (e.g. Pulinets and Boyarchuk, 2004).
The state of the ionosphere is particularly sensitive to the LAIC. Its presence as ionised layer at 50-
1000 km altitude above the Earth's surface is important to detect any electromagnetic change in the
circumterrestrial environment (Kelley, 2009). Comprehensive reviews of the papers describing the
measurements of the seismo-ionospheric signals are reported in De Santis et al. (2015b) and Jin et
al. (2015). In addition, Pulinets and Davidenko (2014) make a discussion on the temporal and
spatial variability of the ionospheric precursor summarizing the results obtained by a large number
of authors so far. In particular, they describe in detail which is the role of the Global Electric Circuit
in transferring information from the Earth's surface up to the ionosphere.
The finding of atmospheric anomalies prior to large EQs is more recent and also debated as well.
In this section, we remind some of those phenomena, the nature and characteristics of which are
more directly of interest for the understanding of LAIC.
*Pre-EQ ionospheric evidences from ground-based observations*
A coupling (post-seismic) effect of an EQ to the above atmosphere is already well known: it can
appear just after the occurrence of a sufficiently large event, and it is related to the possibility of
observing the effect of the propagation of acoustic gravity waves in the ionosphere (e.g. Row, 1967).
Recently, this effect has been clearly detected as wave-like fluctuations of the Total Electron
Content (TEC) in ionosphere 21 minutes after the April 25, 2015 M7.8 Nepal EQ
(http://gpsworld.com/gps-data-show-how-nepal-quake-disturbed-earths-upper-atmosphere/;    last
access on 23 October 2017).
Important precursory effects of LAIC before large EQs can be detected in the ionosphere from
ground-based observational systems like ionosondes and GPS (Global Positioning System)/GNSS
(Global Navigation Satellite System) receivers.
A large number of papers report some variations of ionospheric parameters before many large EQs,
such as the F2-layer critical frequency (foF2) (Hobara and Parrot, 2005; Liu et al., 2006; Dabas et



al., 2007; Perrone et al. 2010; Xu et al. 2015) and the sporadic E layer (Es) (Silina et al. 2001;
Ondoh 2003; Ondoh and Hayakawa 2006).
The study of foF2 alone is a very "inconvenient" ionospheric parameter for the role of EQ precursor,
because, besides the geomagnetic activity effects, there would be many other reasons for non-EQ
related foF2 variations. Therefore, a multi-parameter analysis is preferable and some works have
analysed more ionospheric parameters at the same time, in order to achieve a more robust result.
For instance, in the periods of time preceding all crustal EQs in Central Italy with magnitudes M >
5.0 and the epicenter depth < 50 km, Perrone et al. (2010) have considered the ionospheric sporadic
E layer (Es) together with the blanketing frequency of Es layer (fbEs) and foF2, by analysing data
from the ionospheric observatory inside the preparation zone. According to these authors, the found
deviations of ionospheric parameters from the background level can be related to the magnitude and
the epicentre distance of the corresponding EQ.
There is significant literature related to the analysis of the ionospheric effects before and during an
EQ revealed by GPS/GNSS ground-based measurements, in terms of TEC fluctuations and
scintillation anomalies that have been claimed to be detected some days before the EQs. Just to
mention the more recent works, Mancini et al. (2014) analysed 5 years of GNSS-based ionospheric
TEC data to produce maps over an area surrounding the epicentre of the 2009 L'Aquila EQ. An
interesting ionospheric anomaly was found in the night of 16 March 2009, anticipating the main
shock by 3 weeks and could be connected with it. Contadakis et al. (2014) reported on the analysis
of the total electron content (TEC) from eight GPS stations of the EUREF network by using
discrete Fourier to investigate the TEC variations over the Mediterranean region before and during
the 12 October 2013 Crete, Greece EQ. Over an area of several hundred kilometers from the EQ
epicentre, all stations used in this study observed an increase of 2-6 TECU from 10 October to 15
October 2013, likely related to the EQ. Akhoondzadeh (2015) applies a complex algorithm, the
Firefly Algorithm (FA), as a robust predictor to detect the TEC seismo-ionospheric anomalies
around the time of the some powerful EQs (27 February 2010 M8.8 Chile, 11 August 2012 M6.4
Varzeghan and 16 April 2013 M7.7 Saravan). Significant anomalies were observed 3 - 8 days
before the EQs.
A recent paper by Kuo et al. (2015) presents the application of the LAIC model to compute the TEC
variations and compare the simulation results with TEC observations for the Tohoku-Oki EQ (Japan,
11 March 2011, Mw 9.0). In the simulations, these authors assumed that the stress- associated
current starts ~40 minutes before the EQ, and then linearly increases reaching its maximum



magnitude at the time of the EQ main-shock. Comparisons with experimental values suggest that a
dynamo current density of ~25 nA m$^{-2}$ is required to produce the observed variation of ~3 TECU.
However, it must be noted that the relationship between ionospheric anomalies and electromagnetic
signals generated by the EQ preparation is still controversial and highly debated, as demonstrated
by the high number of papers reporting re-analysis of data and comments aiming to refute evidences
of this correlation. For example, Masci et al. (2015) comment the findings of Heki (2011) and Heki
and Enomoto (2013). After a re-analysis of the data, used by Heki (2011) to demonstrate the
existence of a TEC anomaly 40 minutes before the EQs (2011 Tohoku-Oki and other M>8 EQs),
Masci et al. (2015) conclude that this anomaly is due to an artefact introduced by the choice of the
definition of the reference line adopted in analysing TEC variations.
*Pre-EQ ionospheric evidences from in-situ measurements*
Although many works on the possible pre-EQ effects in the ionosphere were also made with the
early advent of satellites, it was with the DEMETER (Detection of Electro-Magnetic Emissions
Transmitted from EQ Regions; 2004-2010) and CHAMP (CHAllenging Minisatellite Payload;
2000-2010) missions that most of the striking results were obtained.
DEMETER was a French micro-satellite operated by CNES and specifically designed to the
investigation of the Earth ionosphere disturbances due to seismic and volcanic activities. It operated
for more than 6.5 years of scientific mission (2004-2010). The results from the analyses of this
satellite dataset have statistically proved definitively the existence of the LAIC: what is still needed
is to understand the deterministic details. Using the complete DEMETER data set (Píša et al. 2013),
careful statistical studies were performed on the influence of seismic activity on the intensity of low
frequency EM waves in the ionosphere. In the satellite lifetime, several thousands of magnitude
M5+ EQs occurred so constituting the seismic database. In particular the normalized probabilistic
intensity obtained from the night-time electric field data is below the "normal" level, shortly (0 − 4
hours) before the shallow (depth < 40 km) M5+ EQs at 1 − 2 kHz. Clear perturbations are observed
a few hours before the EQs, as another example of "imminent" forecast: they are real, although they
are weak and so far only statistically revealed. No similar effects were observed during the diurnal
hours and for deeper EQs. It is interesting also to note that the spatial scale $R$ of the affected area is
approximately 350 km confirming relatively well the size of the EQ preparation zone estimated
using the *Dobrovolsky et al.* (1979) formula. The main statistical decrease is observed at about 1.7




kHz, corresponding approximately to the cut-off frequency of the first transverse magnetic (TM)
mode of the Earth–ionosphere waveguide during the night-time. An increase of this cut-off
frequency effect would therefore necessarily lead to the decrease of the power spectral density of
electric field fluctuations observed by DEMETER in the appropriate frequency range, meaning a
lower height of the ionosphere above the epicenter of the imminent EQ. As the EM waves
propagating in the Earth-ionosphere wave-guide are mainly whistlers, this means that their
propagation is disturbed above the epicenters of future EQs, instead of a change of their intensities.
Ryu et al. (2014a,b) took advantage of the simultaneous measurements of these two satellites: they
analysed the electron density and temperature, ion density composition and temperature data from
DEMETER ISL (Langmuir Probe), ICE (Electric Field Instrument) and IAP (Plasma Analyzer
Instrument), together with CHAMP PLP data (electron density and temperature) and IONEX maps
of vTEC (vertical TEC) from IGS (International GNSS Service). Their aim was the investigation of
the ionospheric fluctuations related to the EQs occurred in September 2004 near to the south coast
of Honshu, Japan (Ryu et al, 2014a) and Wenchuan EQ (M7.9) of 12 May 2008 (Ryu et al, 2014b).
The main result was the detection of a gradual enhancement of the EIA (Equatorial Ionospheric
Anomaly) intensity starting one month prior to the event, reaching its maximum eight days before,
followed by a decreasing behavior, very likely due to an external electric field generated over the
epicenter affecting the existing $\mathbf{E}{\times}\mathbf{B}$ drifts responsible of the EIA.
By analysing the magnetic data from Swarm satellites of the European Space Agency, a recent
paper (De Santis et al. 2017) finds some important patterns before the April 25, 2015 M7.8 Nepal
EQ, that resemble the same obtained from the seismological analysis of the foreshocks.
*Pre-EQ atmospheric evidences*
The improvement and increase of satellite remote sensing missions go back to early 1980's. Since
then, evidences of many types of infrared (IR) physics parameters have been recognized as useful to
identify possible pre-EQ anomalies. Among them, the most cited are the Brightness Temperature
(BT), Outgoing Longwave Radiation (OLR), Surface Latent Heat Flux (SLHF), Skin Surface
Temperature (SST), and the atmospheric temperature at different altitudes. Although the topic is
still debated or even controversial, many scientists agree that those parameters could change before
EQs and so they are regularly recorded by satellite at regional and global scales. Examples of such
variations of temperature or aerosols can be found in Pulinets et al. (2006), Jing et al. (2013), and
Akhoondzadeh (2015). BT corresponds to the temperature of a black body that emits the same
intensity as measured, and Xie and Ma (2015) found a clear BT anomaly in correspondence of



Lushan M7 EQ (China). OLR is the emission of the terrestrial radiation from the top of the Earth's
atmosphere to the space; it is controlled by the temperature of the earth and the atmosphere above it,
in particular, by the water vapor and the clouds: as examples, Ouzounov et al. (2011) reported
anomalies in this parameter days before the seismic events. SLHF describes the heat released by
phase changes and shows an evident dependence on meteorological parameters such as surface
temperature, relative humidity, wind speed, etcetera. SST is the temperature of the Earth's surface
at radiative equilibrium (usually, the interface between soil and atmosphere, on lands; it is identical
to Sea Surface Temperature over the seas), in contrast with the meteorological definition of surface
temperature measured by air thermometers which take readings at approximately 1 meter above
ground level. We will study the SST for the epicentral areas of the L'Aquila and Emilia main-
shocks.
The nature of the detected IR anomaly as a real temperature change, or perhaps just an emission in
the IR frequency band, is a debated issue. In a recent paper, Piroddi et al. (2014) show a clear
Thermal IR (TIR) anomaly preceding the 2009 M6.2 L'Aquila (Italy) EQ. The authors propose a
mechanism of generation of electric currents in the lithospheric rocks when they are under stress
and a consequent IR irradiation with no actual temperature change (e.g. Freund, 2011). However,
some recent works identified SLHF (Qin et al., 2011) and surface temperature anomalies (Qin et al.,
2012) occurring before large EQs, thus supporting the possibility for some actual change of
temperature too. Although the exact cause of such temperature raise is still unknown, it is possible
to definitely exclude the radon as a possible direct heat source, on the basis of the results of
laboratory experiments conducted by Martinelli et al. (2015). Pulinets et al. (2015) resort to another
role of radon as possible indirect source: it could drive particle ionization and aerosol aggregation,
where the latent heat release can cause the found increase in the atmospheric temperature.
Application of particular sophisticated techniques is mandatory to identify the anomalous signal in
the TIR data. For instance, Tramutoli (2007, 2010), Aliano et al. (2008), and more recently Xiong et
al. (2015) propose some robust satellite techniques that take into proper account the past behaviour
of the signal under investigation: the typical seasonal and yearly background is computed and
statistically significant deviation from it may represent the thermal anomaly. Recent interest was
also addressed to air-quality data as possible indicators of an impending EQ (e.g. Hsu et al., 2010):
these authors found a staggering increase in ambient $SO_2$ concentrations by more than one order of
magnitude across Taiwan several hours prior to two (M6.8 and M7.2) significant EQs in the island.

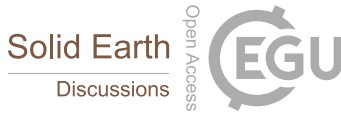



An interesting, although still controversial, emerging study concerns the EQ clouds (Guangmeng
and Jie, 2013), suggesting that their formation is due to some local weather conditions caused by
energy and particle exchanges between crust and atmosphere able to locally modify the global
electric circuit during the EQ preparation phase (e.g. Harrison et al., 2014); or to create the
conditions for electrical discharges in an atmosphere that may be the source of very high frequency
(VHF) radio-emissions, sometime detected prior to large EQs (Ruzhin and Nomicos, 2007).
Recently, the claim of unusual cloud formation prior strong EQs by Guangmeng and Jie (2013) was
strongly questioned by Thomas et al. (2015) with a counter-analysis based on examination of 4
years of satellite images and correlation analyses between linear-cloud formations and EQ
occurrence.
*Physical Models*
A plausible physical omni-comprehensive model justifying the great variety of evidences given
before is the real difficult conundrum for the scientists in this field. There are many theories that
attempt to describe the physical processes manifesting anomalous behaviour in some parameters
before the occurrence of an EQ and try to explain what could cause these precursors. A review of
these processes can be found in Pulinets and Boyarchuk (2004), Freund (2011), Pulinets and
Ouzounov (2011) and the references therein (Figure 7).

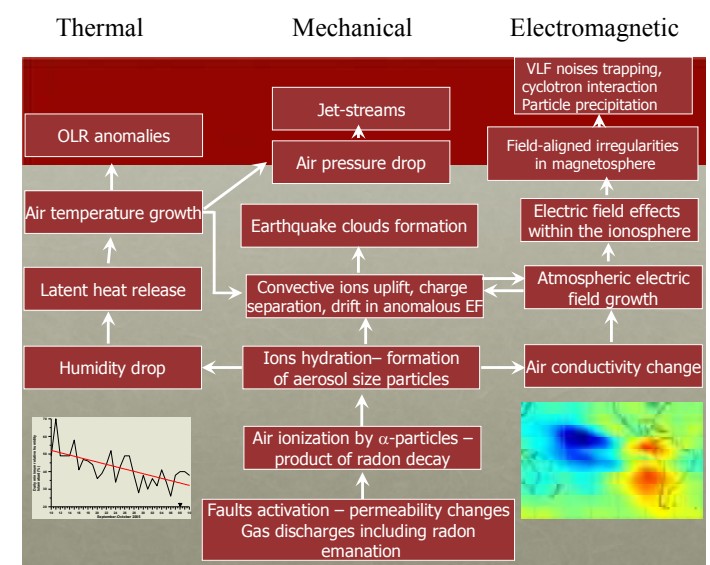

**Fig.7.** Pulinets-Ouzonouv LAIC model (adapted from Pulinets & Ouzonouv, 2011; Pulinets et al., 2015)





Among the many proposed mechanism of generations we can generally classify them as mechanical
(atmospheric waves generated by earth motions) and electrical (electric fields in Earth's crust)
sources: among the former, we can count the various kinds of atmospheric waves as internal or
acoustic gravity waves (IGW and AGW, respectively), planetary waves and tides. In particular, the
hypothesis of acoustic gravity waves generation before EQs was proposed by many authors (e.g.
Pulinets and Boyarchuk, 2004).
More complex and intriguing are the mechanisms that describe the anomalous electric field
generation. A theory that can explain many observations is based on the emission of a radioactive
gas or metallic ions before an EQ, which may change the distribution of electric potential above the
surface of the Earth and then up to the ionosphere (e.g., Sorokin et al., 2001).
Whatever its source is, penetration of the electric field into the ionosphere could induce anomalies
in the ionospheric plasma density and/or conductivity, which are observed above seismic zones (see
e.g., Liu et al., 2006; Kon et al., 2011). In contrast with this view, Harrison et al. (2010) proposed
that radon emitted before an EQ would increase the conductivity of air at ground level and that the
ensuing increase of current in the fair weather global circuit would lower the ionosphere. This
mechanism is also supported by Pulinets et al. (2015). However, Freund et al. (2009) have
estimated that even if radon is coming out the ground in seismic areas, its contribution to the air
conductivity is of minor importance relative to the air ionization rate, which can be expected from
charge carriers from the rocks, the so-called positive-holes (or p-holes) (Figure 8).

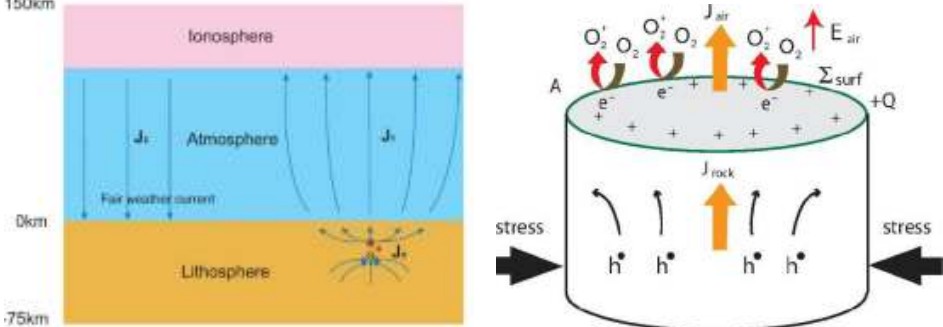

**Fig. 8.** Freund model (adapted from Kuo et al., 2014)



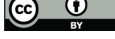

They have shown experimentally that these mobile electric charge carriers flow out of the stressed
rocks (see Freund et al., 2009, and references therein) and, at the Earth's surface, they cause extra
ionization of the air molecules. However, the original experiments that detected these p-holes have
been recently contrasted (Dahlgren et al., 2014; but see also Scoville et al., 2015).

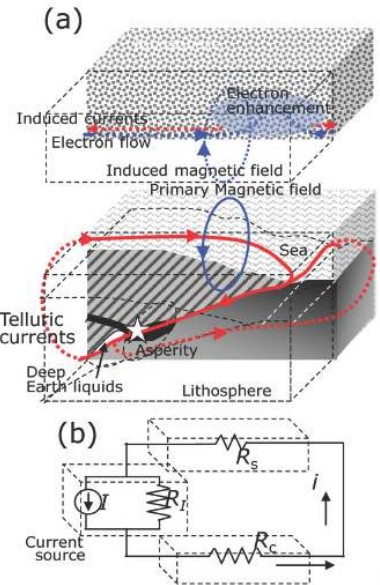

**Fig. 9.** Enomoto model (adapted from Enomoto, 2012)
Kuo et al. (2011, 2014) have shown that ionospheric density variations can be induced by changes
of the current in the global electric circuit between the bottom of the ionosphere and the Earth's
surface where electric charges associated with stressed rocks can appear. The interaction of the
anomalous electric current with the geomagnetic field can even amplify the effect in the higher
atmosphere (Kuo et al., 2014).
Enomoto (2012; Figure 9) has introduced a fault model that takes account of the couple interaction
between EQ nucleation and deep Earth gases, and proposes a physical model of magnetic induction
coupling with ionosphere before large offshore EQs.
**11. Examples of thermal coupling before L'Aquila and Emilia EQs**
In the LAIC model, an important feature should be the coupling between lithosphere and low
atmosphere (i.e. the troposphere) in terms of a thermal coupling. As said in the previous section, no
general consensus exists about the fact that the thermal anomaly is just an infrared effect (e.g.
Freund, 2011) or a real change of temperature (e.g. Qin et al., 2012). We do not want to express



here a clear position in this debate. Rather, as didactical examples, we will show some SST studies
for the same cases we analysed for the entropy, i.e. the 2009 L'Aquila and the 2012 Emilia
sequences of EQs.
In each case study, we will consider the SST in the epicentral region about two months around the
EQ occurrence, and then we will compare the temperatures with those measured in the same day, at
the same time (06:00UT) in the time interval 1979-2008 (2011) for L'Aquila (Emilia) EQ. An
anomaly of the physical quantity of concern is defined as a value that exceeds the mean (or median)
by two times the standard deviation, and persists for at least two days (see also Piscini et al., 2017).

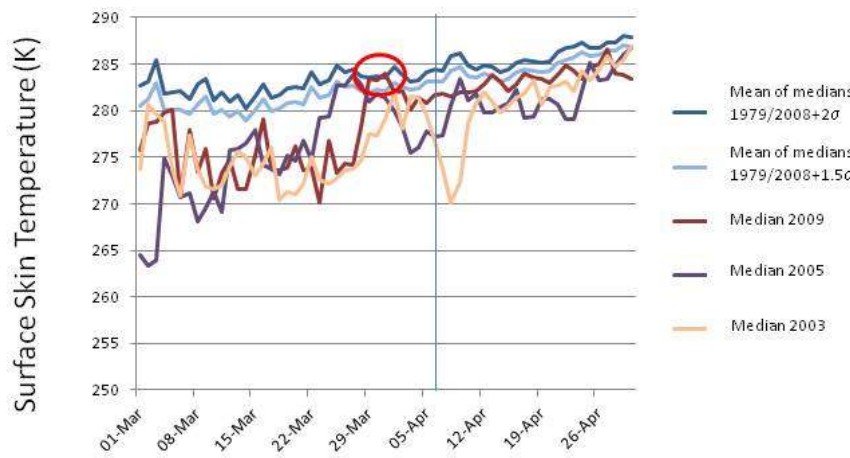

**Fig. 10.** Median behaviour of 2009 from 1 March to 30 April, compared with all 1979-2008 medians, and particular
comparison with 2003 and 2005 medians. All values have been estimated at the epicentre. The red oval indicates when
the thermal anomaly in 2009 is larger than or equal to 2 standard deviation, σ (as computed from the previous 1979-
2008 years) and persists for at least two days. The vertical line is the EQ occurrence.
Fig. 10 and Fig. 11 show the results for the two analyses. In detail, Fig.10 (Fig.11) shows for
L'Aquila (Emilia) EQ the median behaviour of 2009, from 1 March (April) to 30 April (31 May),
compared with all 1979-2008 (2011) medians, and particular comparison with 2003 (2004) and
2005 (2006) medians. For each day the use of the median was preferred because it was thought to
be a more robust indicator. The latter years have been used for comparison because no significant
seismicity occurred in those years in the two considered regions. All values have been estimated at
the EQ epicentre. The red oval indicates when the thermal anomaly in 2009 (2012) is larger than or



equal to 2 standard deviations, σ (as computed for each day from the previous 1979-2008 (2011)
years) and persists for at least two days. In both analyses, a clear anomaly is found around a week
before the EQ occurrence (vertical line in both figures). In the case of Emilia EQ, another persisting
anomaly is also found around 1 month and half before the main-shock.

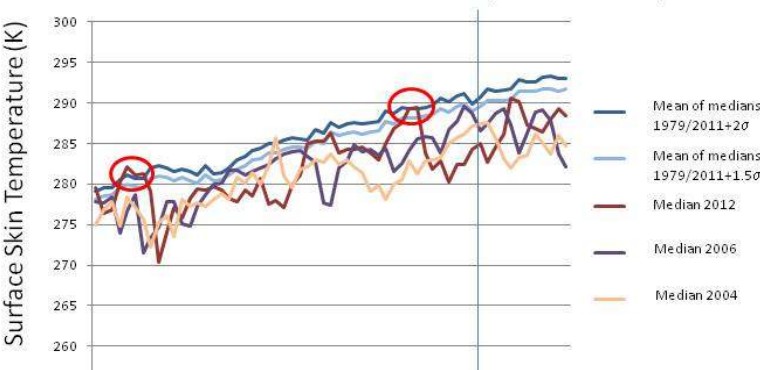

**Fig. 11.** Median behaviour of 2009 from 1 April to 31 May, compared with all 1979-2011 medians, and particular
comparison with 2003 and 2005 medians. All values have been estimated at the epicentre. The red ovals indicate when
the thermal anomaly in 2012 is larger than or equal to 2 standard deviation, σ (as computed from the previous 1979-
2012 years) and persists for at least two days. The vertical line is the EQ occurrence.
These results confirm some previous studies on the possible thermal coupling in the two EQ cases
(e.g. Piroddi et al., 2014; Qin et al., 2012). The same Central Italy showed an analogous thermal
anomaly around 40 days before the recent 24 August 2016 M6 Amatrice EQ: Piscini et al. (2017)
applied the CAPRI algorithm (CAPRI stands for "Climatological Analysis for seismic PRecursor
Identification") that removes the long-term trend over the whole day by day dataset. This procedure
is used mainly to remove a possible "global warming" effect, avoiding to classify as abnormal a
more recent year just because of global warming. These authors integrated the analysis of the skin
temperature also with total column water vapour and total column of ozone and made a confusion
matrix analysis for the last twenty years.





**12. Mutual Information and Transfer Information: a possible future direction**
Geosystemics focuses on the inter-relations among the components composing the terrestrial
complex system. For this reason, every statistical (or physical) quantity that measures these inter-
relations is useful. However, given that the system under study is not usually linear, instead of
linear quantities such as correlation coefficient or cross-correlation function between two variables
belonging to linear processes, we have to resort to statistical quantities, which are more appropriate
for nonlinear processes, as typical in a complex system.
Given two variables X and Y, characterising two processes of the phenomenon under study, we
define the mutual information I(X, Y) extending definition (1) to two variables, i.e.:

$$I(\mathrm{X};Y) = \sum_{y \in Y} \sum_{x \in X} p\ (x,y) \cdot \log\left(\frac{p(x,y)}{p_1(x)p_2(y)}\right) \tag{10}$$

where $p_1(x)$ and $p_2(y)$ are the corresponding probabilities and p($x,y$) is the joint probability.
However, this formulation does not provide hints about the direction of information transfer
between process X and process Y, i.e. from a part of a system to another. For this purpose, it is
possible to introduce a useful definition that quantifies the information flow in terms of the
Kullback and Leibler entropy (Kullback and Leibler, 1951), which can be defined for a single
process X as:

$$K_x = \sum_x p(x) \cdot \log[p(x)/q(x)] \tag{11}$$

The above quantity is the entropy related to the process X when a different probability $q(x)$ is used
instead of the true $p(x)$, and then adapt it in order to be applied to two variables taking into account
of a proper delay in one (Schreiber 2000). The so-called transfer information (changing its sign it
becomes the transfer entropy) provides also the direction of the information flow.
Here, we do not describe more details but we just want to emphasise the importance of quantifying
direction of information flow amongst different parts or processes of the system under study,
because often it is more important to know where the flow of information is going, instead of just
estimating the information that is exchanged by the whole process between internal components or
external ones (Ahlswede et al. 2006).
Applying this concept to two different time series can be useful to say if one is the master quantity
(that represents the causal process) and the other the affected one. In all cases, where we would like



to compare/correlate a seismic sequence with possible atmospheric or ionospheric series of
precursors, the calculation of the transfer information would provide a robust answer.
**13. Conclusions**
This paper has introduced the concepts of geosystemics, and then has shown its applications to
some case studies. The spirit of geosystemics is to use some universal tools to look at some
macroscopic quantities, such as the entropy, the Benioff strain, or the temperature to consequently
deduce macroscopic properties of the physical system under study. An important frame is that of
dynamical systems approaching a critical point, when the macroscopic properties of the system
change dramatically. This could be the case of a sequence of EQs that culminates with a main-
shock. Therefore, we showed some results obtained with the study of two recent Italian seismic
sequences, the 2009 L'Aquila and the 2012 Emilia sequences.
It is obvious that, being the study of EQs a very complex problem, the more characterizing
parameters are analysed, the more robust the result will be. A recent and extensive example of this
approach is given by Wu et al. (2016) for the case of the 2009 L'Aquila seismic sequence.
A further question is how we can use the Big Data in geosciences and, in particular, to analyze
precursory patterns of big earthquakes. Of course, the analysis of a greater number of data and the
check of multiple models is perceptible that allows to find some type of pattern before an
earthquake, that could be likely valid only for regions very localized. An extensive statistical big
data analysis would be important to confirm or confute the individual case results. An example of
this approach is given by Piscini et al. (2017) where the validity of local climatological variations as
possible seismic precursors in Central Italy has been statistically established.





**Acknowledgements**
The authors thank ESA (European Space Agency) for funding the SAFE (**Sw**a**rm f**or **E**arthquake
study) project under STSE Swarm+Innovation Programme, and INGV (Istituto Nazionale di
Geofisica e Vulcanologia) for funding LAIC-U (**L**ithosphere-**A**tmosphere-**I**onosphere **C**oupling
**U**nderstanding) and ECHO (**E**ntropy and **CH**a**O**s: tools for studying and characterizing seismic
sequences evolution) Projects, under which much work was undertaken for preparing this work.



**Appendix:  Derivation of Eq. (6)**
The derivation of eq. (6) is as follows:
The probability density function corresponding to the Gutenberg-Richter law is

$$p(M) = \frac{b \cdot 10^{-b(M-M_{\min})}}{\log e} \quad \text{with } M \geq M_{\min}$$
(A1)

and, imposing $M_{min} = M_c$, hence:

$$H = -\int_{M_c}^{\infty} p(M) \cdot \log p(M) dM = -\int_{M_c}^{\infty} \frac{b \cdot 10^{-b(M-M_c)}}{\log e} \log\left(\frac{b \cdot 10^{-b(M-M_c)}}{\log e}\right) dM =$$

$$= -\frac{b}{\log e} \int_{M_c}^{\infty} 10^{-b(M-M_c)} \left[\log b - b(M - Mc) - \log(\log e)\right] dM$$

$$= -\frac{b}{\log e}\left\{\left[\log b - \log(\log e)\right]\int_{M_c}^{\infty} 10^{-b(M-M_c)} dM - b\int_{M_c}^{\infty}(M - M_c)\,10^{-b(M-M_c)}\,dM\right\}$$

$$= -\frac{b}{\log e}\left\{\left[\log b - \log(\log e)\right]\frac{\log e}{b} - b\frac{(\log e)^2}{b^2}\right\} = -\log b + \log(\log e) + \log e,$$
A(2)

14  which agrees with equation (6).





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
