# Peer review of "Discussion started: 30 January 2018 © Author(s) 2018. CC BY 4.0 License."

_Solid Earth, 2017_

## Referee Comment (RC1) · Anonymous Referee #1 · 31 Jan 2018

Dear editor

I have reviewed this (mostly unchanged) manuscript last year for a different journal and recommended rejection due to significant problems. I have outlined the problems in my review that is attached. A quick glance shows that the authors have changed their introduction but the scientific description is basically unchanged although I have pointed out large problems.

Therefore, I am not willing to spend more time on reviewing another submission of the manuscript if the authors are not willing to spend their time on improving their manuscript. Receiving reviews from others and ignoring them is unfair to others who have put work into reading and reviewing the authors' work. It is undermining the review process if the content of reviews is completely ignored. Why should reviewers spend their time on reporting problems in detail if authors choose to simply resubmit to

another journal without even taking the critique into account?

Furthermore, as can be seen in my attached review, I have pointed out how earthquake precursor studies should be conducted and that basically each of the outlined five steps is missing in this manuscript. The irony in this resubmission is that the authors have included this outline in their manuscript (5-point bullet list on page 6) without addressing the five points at all.

In conclusion, I can, again, only recommend rejection of this manuscript and urge the authors to rethink their manuscript in the light of my and possibly others' reviews.

Please also note the supplement to this comment:
https://www.solid-earth-discuss.net/se-2017-120/se-2017-120-RC1-supplement.pdf

**Supplement:**

**Review of**
**"Geosystemics and Earthquakes"**
**by**
**Angelo De Santis, G. Cianchini,**
**R. Di Giovambattista, C. Abbattista, L. Alfonsi,**
**L. Amoruso, M. Carbone, C. Cesaroni,**
**G. De Franceschi, Anna De Santis, D. Marchetti,**
**F. J. Pavòn-Carrasco, L. Perrone, A. Piscini,**
**L. Spogli and F. Santoro**

This paper claims to be a review of a field the authors call *Geosystemics*. However, it lacks what I would expect from a review: An unbiased introduction to all relevant works in the particular field and a sober evaluation of their quality and importance. Instead, the manuscript is an enthusiastic description of various works from the 'earthquake-prediction' community. This community is widely critized for not following standards in investigating earthquake predictions, as I will explain below. Furthermore, this so-called review is by no means a review as it focuses solely on the publications from the investigating authors and mainly ignores critical works. Last but not least, this manuscript is certainly not a review of what the authors call *Geosystemics* but rather an advertment for the 'earthquake-prediction' community and the lead author (14 references point to papers of the lead author). I regret but I cannot recommend this manuscript for publication.

Besides the problem that this manuscript does not constitute a review, the topic itself is very problematic. Even though the authors mention the Collaboratory for the Study of Earthquake Predictability (CSEP) with a web link, they do not consider the work of CSEP at all and ignore the

standards that CSEP has set for testing earthquake forecasts or predictions and that should be followed by any publication of earthquake precursors:

1. *Present a physical model that can explain the proposed precursor anomaly.*

   To some degree, the authors present physical ideas for the precursor phenomena discussed. However, several precursory phenomena are introduced without any physical reasoning why they should work. Particularly in light of a review paper about *Geosystemics*, this lack of descriptions remains puzzling. I am not arguing that there cannot be a precursory phenomenon that is not understood but may show some predictive power, however, the presented ideas have so far failed to show their predictive power in prospective and independent tests and, therefore, their simple description does not suffice.

2. *Exactly define the anomaly and describe how it can be observed.*

   The entire paper is filled with observations and their respective interpretations. However, the authors do not define any of the anomalies in a testable way. Particularly outstanding examples of such 'anomalies' are shown in Figures 10 and 11. This type of anomalies is typical in publications of the 'earthquake-prediction' community as they suggest an anomalous signal without defining this anomaly and showing that it appears only (or at least mostly) before large earthquakes. This type of evidence always bears the problem of selection bias and it is the duty of the authors to show their readers that they did not fall for selection bias. Without a clear and quantitative description of an anomaly, all these observations are basically meaningless as they stem from observations in hindsight and prone to selection bias. It is a characteristic of such studies that the search for anomalies is conducted after large earthquakes have been observed. Usually, the parameters of the applied metric are tweaked until the anomaly is found. Thus, the authors need to fully pre-specify what the metrics for each anomaly exactly are and then apply it to future data to see if the anomaly idea holds.

3. *Explain how a precursory information can be translated into a forecast and specify such a forecast in terms of probabilities for given space/time/magnitude windows.*

   This aspect is completely lacking in the manuscript. And it is also lacking in the presented methods. Only the PI method was ever tested independently in the framework of CSEP. The RTP method has at least published their forecasts but the results were not convincing. From all the other methods I have never seen a testable model or any collaboration with CSEP. Quite the contrary, the CSEP team has tried several times to initiate collaborations but without success. I assume the lack of a testable model may have been the reason behind this.

4. *Perform a test over some time that allows to evaluate the proposed precursor and its forecasting power.*

   This is the gold standard for seismic precursor studies. Even though many precursors have been proposed a long time ago, no such tests have ever been carried out for the methods described in this manuscript. The sole execption is the PI method for which a forecast was submitted in 2006 and has been tested for the past 10 years, revealing no particular forecasting power when compared to the other forecasts in the test [*Schorlemmer et al.*, 2010; *Strader et al.*, 2017]. However, some of the authors of the PI method have published their own evaluation of the tests by changing the definitions of the metrics that were agreed upon by all participants before the tests took place [*Lee et al.*, 2011]. Not surprisingly, the PI showed higher forecasting power in their evaluation. For most of the other methods, only investigation in hindsight after large earthquake took place which do not constitute systematic testing.

5. *Report on successful prediction, missed earthquakes, and false predictions.*

   Limiting the search to periods shortly before large observed earthquakes will never be sufficient to investigate the predictive power of the precursory phenomena. The authors need to expand their search systematically and report on all hits, misses, and false alarms.

I could certainly formulate more critical points about the presented methods and why their approach to percursory phenomena is not sufficiently scientific, however, I consider it out of the scope of this article. Nevertheless, even if the authors are not responsible for non-scientific work of others, they have the duty to report on the problems, in particular the scientific problems, of the presented and introduced methods. Without it, this so-called review remains a biased advertisement for the 'earthquake-prediction' community and does not provide the means for an unbiased understanding of precursory phenomena. In light of this, the notion of *Geosystemics* seems to be very limited to the authors' works.

**References**

Lee, Y.-T., D. L. Turcotte, J. R. Holliday, M. K. Sachs, J. B. Rundle, C.-C. Chen, and K. F. Tiampo, Results of the Regional Earthquake Likelihood Models (RELM) test of earthquake forecasts in California, *PNAS*, *108*(40), 16,533–16,538, doi:10.1073/pnas.1113481108, 2011.

Schorlemmer, D., J. D. Zechar, M. J. Werner, E. H. Field, D. D. Jackson, T. H. Jordan, and the RELM Working Group, First Results of the Regional Earthquake Likelihood Models Experiment, *Pure and Applied Geophysics*, doi:10.1007/s00024-010-0081-5, 2010.

Strader, A., M. Schneider, and D. Schorlemmer, Prospective and retrospective evaluation of five-year earthquake forecast models for California, *Geophys. J. Int.*, *211*(1), 239–251, doi:10.1093/gji/ggx268, 2017.

---

## Short Comment (SC1) · 31 Jan 2018

This is an excellent paper, which discusses the wide range of observables that have been reported for pre-earthquake conditions. Congratulations to the authors.

on p. 25, L 13-14 the authors touch upon a question that has been discussed for many years, namely the physical cause of the IR emission. The paragraph begins with the words "The nature of the detected IR anomaly as a real temperature change, or perhaps just an emission in the IR frequency band, is a debated issue." They continue with the statement that there might be "IR irradiation with no actual temperature change (e.g. Freund, 2011)" and contrast this with "some recent works identified SLHF (Qin et al., 2011) and surface temperature anomalies (Qin et al., 19 2012) occurring before large EQs, thus supporting the possibility for some actual change of temperature too."

From my perspective, there is no contradiction. Here is why.

[Figure]

Figure 1 illustrates the basic scenario that, deep in the Earth's crust, where peroxy defects (typically O3Si-OO-SiO3) abundantly exist in igneous and high-grade meta-morphic rocks, peroxy defects become activated. As the peroxy bonds break, they release two types of electronic charge carriers, electrons e' and holes h•.

The e' are confined to the stressed rock volume, but the h• have the ability to flow out of the stressed rock volume, spreading fast (on the order of 100 m/sec) and far (on the order of kilometers to tens of kilometers) into the surrounding rocks. The h• are electronic charge carriers, in semiconductor parlance "defect electrons in the oxygen anion sublattice". They propagate as electronic charge states. No mass transport. Since they are positively charged, they repel each other. When they arrive at the Earth's surface, they tend to go to the topographic highs such as hill tops and mountain tops. There they become trapped at the ground-to-air interface. The more h• arrive, the higher the number density per unit surface area.

Eventually, at sufficiently high number densities, the h• will begin to recombine, forming new O3Si-OO-SiO3 bonds as depicted in the top left of Figure 1. The recombination is an exothermal reaction, yielding about 2.1 eV energy. This energy is channelled into the two oxygens (valence 1-) that form the peroxy bond. They become vibrationally highly excited to the tune of ~20,000 degrees Kelvin equivalent. During de-excitation, the newly formed peroxy bonds emit IR radiation as depicted in the top right of Figure 1. The IR photons are emitted at discrete energies corresponding to the downward quantum transitions of the O-O bond in its vibrational manifold. This IR emission con-sists of a series of sharp bands in the spectral region 10-14 µm plus "pink noise" over the same spectral region and beyond towards even longer wavelengths.

We have experimentally obtained more than 10 of these IR emission bands (Freund and Scoville, unpubl.) We used them to derive the Morse potential of the O3Si-O<>O-SiO3 bond and found perfect agreement with high-level quantum mechanical calcu-lations of the O3Si-OO-SiO3 defect in silica glass (Ricci et al. "Modeling disorder in amorphous silica with embedded clusters: the peroxy bridge defect center" Phys. Rev.

B, 64, 224104 1-8, 2001).

However, the success of this determination should not close our minds toward the fact that any stimulated IR emission from vibrationally very "hot" systems is not a "clean" process. Instead of just falling down the quantum ladder within the peroxy vibrational manifold, energy is being channelled sidewise to other atoms in the peroxy neighborhood, causing them to become vibrationally excited. They will then emit their own IR photons. Eventually, the system will "thermalize", meaning literally that each newly formed peroxy bond on the surface of the Earth will become a "hot spot", surrounded by a small halo where the neighboring atoms have actually increased their Joule temperature.

Hence, there is no contradiction between the co-occurrence of stimulated IR emission and the "warming" of the Earth surface – no contradiction between "hot" IR emission and "the possibility for some actual change of temperature too" as quoted in the De Santis et al. paper. It's all part of the same physical process.

———————————————

[Figure]

[Figure]

**Figure 1** Schematic representation of the basic processes involved in the stress activation of peroxy defects in a rock volume that will become the hypocenter of an earthquake, the propagation of h• charge carriers and their trapping at the Earth's surface, their exothermal recombination to new peroxy bonds, and the stimulated emission of IR photons of specific wavelengths (Freund unpubl).

**Fig. 1.** IR emission by radiative de-excitation of peroxy bonds

---

## Author Comment (AC1) · 6 Mar 2018

First of all, we really thank Dr. Freund for the very positive comments about our paper. He recognizes our efforts to combine a wide range of observables for a deep investigation of the pre-earthquake conditions. He also clearly understands the difficulty to proceed along this path (see the prejudiced view by the other referee).

Second, we appreciate the clear explanation for the apparent paradox related whether the IR effects are with thermal effects.

To insert Dr. Freund clarification in our revised manuscript, we added the following statement in pag.25:

"According to Freund (2018), this is only an apparent paradox because any stimulated IR emission from vibrationally very "hot" systems is not a "clean" process. Eventually,

the system will "thermalize", meaning literally that each newly formed peroxy bond on the surface of the Earth will become a "hot spot", surrounded by a small halo where the neighboring atoms have actually increased their Joule temperature."

as a short extract of the Dr. Freund's comment itself , which is also directly referred, so any potential reader can directly read the more complete Freund arguments.

Please also note the supplement to this comment:
https://www.solid-earth-discuss.net/se-2017-120/se-2017-120-AC1-supplement.pdf

---

## Author Comment (AC2) · 6 Mar 2018

We thank the anonymous referee for the criticisms that give us the chance to underline once more (see below), despite his/her negativity, the high level of contribution of this investigation to the worldwide scientific debate and efforts in understanding the earthquake preparation phase in order to arm the scientific community and stakeholders against the natural disasters. By the way, we now stress this concept also in the conclusions of the revised manuscript.

As a general reply to the referee, we would like to highlight the following: When he/she says: "Last but not least, this manuscript is certainly not a review of what the authors call Geosystemics but rather an advertisement for the 'earthquake-prediction' community and the lead author (14 references point to papers of the lead author)." In contrast with referee's point of view, we believe that our paper is a good source of information

regarding the geosystemics and its possible application to earthquakes: in particular, it provides 117 references about the subject. In addition, please note, that less than 12% of references dedicated to papers of the lead author is not a great portion of all quoted references (14 out of the total 117). Our own references were useful in order not to repeat what made with more detail in that specific cited literature.

Regarding the various points as deduced from CSEP that we list at pag.6, we reply as follow:

First, all CSEP points are sequential, i.e. any precursor must sequentially satisfy those points. If a precursor is at an initial stage of research, it satisfies not all points but only some of the first, and this is the case of the most recent found precursors we presented (e.g. entropy). Generally speaking we would appreciate the possibility to describe single case studies where some anomaly appear before the occurrence of earthquakes, giving to the scientific community the possibility for future deeper investigations.

Now we pass to examine each of the CSEP points, about which this Referee says we did not consider at all.

1. This point concerns the presentation of a physical model for the precursor phenomena. Although the reviewer admits that we "introduced" several precursory phenomena, he says that we did it without any physical reasoning. This is not true. Section 10 is completely dedicated to LAIC (pag.20-28) and particular attention to physical models in pages 26-28.

2. About the definition of the anomaly. The reviewer says that we do not define any of the anomalies in a testable way, and present as "particularly outstanding examples of such anomalies" those in Figures 10 and 11. Probably the reviewer did not read carefully the paper when we define it, very generally: "we intend here for an "event" as an anomalous behaviour of the system evolution, e.g. when its signal level is larger than a certain number of standard deviation, $\sigma$, e.g. 2.5 $\sigma$" (lines 17-19, pag.11) and, then, more clearly: "An anomaly of the physical quantity of concern is defined as a value that exceeds the mean (or median) by two times the standard deviation, and persists for at least two days (see also Piscini et al., 2017)." (lines 7-9, pag.29). In the lines above we also define the set of data over which estimate mean (or median) and standard deviation: "In each case study, we will consider the SST in the epicentral region about two months around the EQ occurrence, and then we will compare the temperatures with those measured in the same day, at the same time (06:00UT) in the time interval 1979-2008 (2011) for L'Aquila (Emilia) EQ." (lines 5-7, pag.29). By the way, the latter two sentences were made for explaining Figures 10 and 11.

3. and 5. although are not met in the paper, we refer a paper that copes clearly with these points (Piscini et al., 2017), by means of a confusion matrix of the overall results.

4. As this referee says: "This is the gold standard for seismic precursor studies". This is the most difficult part of the all points, where a prediction is performed on the basis of future results. It would require probably years before a precursor can be properly and fully tested, so it cannot performed for a precursor study at its early stage.

For the above considerations, we added a paragraph in our manuscript just after the mentioning of the 5 points of CSEP.

Please also note the supplement to this comment:
https://www.solid-earth-discuss.net/se-2017-120/se-2017-120-AC2-supplement.pdf

**Supplement:**

[revised manuscript text omitted]

All above points are sequential, i.e. any mature precursor must sequentially satisfy those points. However, if a precursor is at an initial stage of maturity, for instance, it has been just discovered in some case studies, it can satisfy only some of the first points, lacking the following ones. An early stage of the work on some novel precursors cannot exclude the publication of initial investigations. This is the case of most recent found precursors (e.g. entropy) that we will present below.

In the present paper we surely meet the first two points, leaving the other three points to other papers where a deeper and extended study is performed on a few precursors (e.g. Piscini et al. 2017)

[revised manuscript text omitted]

---

## Referee Comment (RC2) · Anonymous Referee #2 · 16 Mar 2018

This article lists several possible candidates of possible earthquake precursors, including M8, RTP, PI, Shannon Entropy (information), R-AMR, LAIC, with focuses on the latter three, and suggest using mutual information as the index of correlation between these phenomena and earthquakes.

The paper has been clearly written, but the shortcoming of this study is also clear. As written in this article (Lines 4 to 18, Page 6), statistical seismology has been developed for many years and provides us with tools for evaluating predictive powers of earthquake predictions and forecasts. It is possible for the authors to evaluate their precursor candidates in rigorous statistical tests. But such tests are absent from this article. Especially, the entropy studies can be applied systematically to a larger catalog, like the SCEDC catalog or the JMA catalog, to evaluate its overall performance, while the author only apply it to 2 cases in Italian catalog. It is hardly believable that we can draw important conclusions only based on these two cases.

[Figure]

Another important issue is that in Sections 8 and 9, the completeness of earthquake catalogs is not considered for the given magnitude thresholds.

Minors.

Lines 13-22, Page 7. RTP has already evaluated by the gambling score, showing only marginal or no significance in predicting Eqks.

Lines 21-27, Pages 32. Please note that big data cannot do everything. Within the big data, there are many pairs of things that have statistical correlation by chance, but not causalities between them.
* * *

---

## Author Comment (AC3) · 23 Apr 2018

This article lists several possible candidates of possible earthquake precursors, including M8, RTP, PI, Shannon Entropy (information), R-AMR, LAIC, with focuses on the latter three, and suggest using mutual information as the index of correlation between these phenomena and earthquakes.

The paper has been clearly written, but the shortcoming of this study is also clear.

We thank this Referee since he indicated some weak points of our work. However we added or changed some parts of the text in order to meet this Referee requests.

As written in this article (Lines 4 to 18, Page 6), statistical seismology has been developed for many years and provides us with tools for evaluating predictive powers of earthquake predictions and forecasts. It is possible for the authors to evaluate their precursor candidates in rigorous statistical tests. But such tests are absent from this article.

We do not agree for two reasons: 1) we refer to the results of a previous paper of ours (Piscini et al. 2017) where a systematical statistical analysis is performed for Central Italy for assess the quality of the skt as possible seismic precursor. To better underline this fact, we now extended this part including also a Table (Table 2). 2) The seismic sequences of 2009 L'Aquila and 2012 Emilia earthquakes are two out of four seismic sequences occurred in Italy in the last 15 years, so they are very representative of the statistics, although the total cases are not so many. By the way, in a previous paper (De Santis et al. 2015) we even analysed a third case (2010 Pollino sequence) so finally analyzing three out of four possible cases. We added a sentence explaining this at the end of section 10.

Especially, the entropy studies can be applied systematically to a larger catalog, like the SCEDC catalog or the JMA catalog, to evaluate its overall performance, while the author only apply it to 2 cases in Italian catalog. It is hardly believable that we can draw important conclusions only based on these two cases.

It is true that we applied the entropy analysis to only two case studies, but in most occasions, we could extend the past results found in literature analyzing *b*-value to the entropy, via the equation (6). We mention this remark at the end of section 6 and 8, added also a reference (Sugan et al. 2014).

Another important issue is that in Sections 8 and 9, the completeness of earthquake catalogs is not considered for the given magnitude thresholds.

We forgot to mention that we actually checked the completeness of the earthquake catalog in both cases. We added a sentence in section 8 and in the caption of Table 1.

Minors.

Lines 13-22, Page 7. RTP has already evaluated by the gambling score, showing only marginal or no significance in predicting Eqks.

We do not agree. This affirmation by Ref. #2 is probably taken after Zechar & Zhuang, 2010. However, according to Molchan, 2011: "The statistical analysis of any prediction method with few target events and a short monitoring period is premature (this is the case of RTP)". We added this in the text and added the two references.

Lines 21-27, Pages 32. Please note that big data cannot do everything. Within the big data, there are many pairs of things that have statistical correlation by chance, but not causalities between them.

This is true. We added a warning in the Conclusions.

**References**

Molchan G. and L. Romashkova, Gambling score in earthquake prediction analysis, Geoph. J. Int., 184, 1445–1454, 2011.

Sugan, M., A. Kato, H. Miyake, S. Nakagawa, and A. Vuan, The preparatory phase of the 2009 Mw 6.3 L'Aquila earthquake by improving the detection capability of low-magnitude foreshocks, Geophys. Res. Lett., 41, 6137–6144, doi:10.1002/2014GL061199, 2014.

Zechar, J.D. & Zhuang, J., Risk and return: evaluating RTP earthquake predictions, Geophys. J. Int., 182, 1319–1326. 2010.

---

## Author Comment (AC4) · 23 Apr 2018

Final Comment

We already responded to Dr. Freund and to Referee #1 on 6 March 2018 (see https://www.solid-earth-discuss.net/se-2017-120/). Now we also included a Reply to Referee #2 (sse our AC3). For Editor's convenience, we summarise all points here.

Dr. Freund was extremely positive while Referee #1 was completely negative. We inserted the main comment by Dr. Freund in the text, so clarifying the point arose by him. We replied to Comments by Referee #1 that we find very skeptical, and probably against any earthquake prediction work. To what already replied in that occasion (see https://www.solid-earth-discuss.net/se-2017-120/) we also can now add what Molchan (2011) says about the systematic analysis to establish whether a method for earthquake prediction is valid or not: "The statistical analysis of any prediction method with few target events and a short monitoring period is premature". Therefore, what we present in our paper in terms of entropy and R-AMR analyses, together with the climatological analysis before large earthquakes is valid and worth doing.

Referee #2, although partially negative, arose some important points that we admit were important.

Our final revision, with adding several parts, a Table and three new references, (here included with all changes evidenced in revision mode) attempts to respond to all these points, and we are confident to have improved the paper significantly, so as the paper is now worth publishing.

Best regards

Angelo De Santis On behalf of all authors

Please also note the supplement to this comment:
https://www.solid-earth-discuss.net/se-2017-120/se-2017-120-AC4-supplement.pdf

[Figure]

**Supplement:**

[revised manuscript text omitted]

All above points are sequential, i.e. any mature precursor must sequentially satisfy those points. However, if a precursor is at an initial stage of maturity, for instance, it has been just discovered in some case studies, it can satisfy only some of the first points, lacking the following ones. An early stage of the work on some novel precursors cannot exclude the publication of initial investigations. This is the case of most recent found precursors (e.g. entropy) that we will present below.

In the present paper we surely meet the first two points, leaving the other three points to other papers where a deeper and extended study is performed on a few precursors (e.g. Piscini et al. 2017).

In this part, we will focus our attention to the deterministic methods, which are essentially based on a systematic catalog-based search of some peculiar *seismicity pattern recognition* in the given area of interest. A wide review on this topic is presented by Mignan (2008). In the following, we will describe *M8*, *RTP* (Reverse Tracing of Precursors), *PI* (Pattern Informatics) and *R-AMR* (Revised Accelerating Moment Release). The latter method is the most recent and is the one we know much better, because some of the present co-authors have introduced the corresponding technique (De Santis et al., 2015b). For this reason, we will dedicate a specific section to it.

*M8*

*M*8 was called in this way because it was designed by retroactive analysis of the seismicity preceding the greatest (M8+) EQs worldwide (e.g. Keilis-Borok and Kossobokov, 1990; Kossobokov, 2013). Some spatio-temporal functions are introduced in order to reconstruct a 7-dimensional phase space (the first three quantities are estimated for two values *C* of the average annual number of earthquakes in the sequence; usually C=10 and 20): *N*(t) is the number of main-shocks; *L*(t) the deviation of *N*(t) from the long-term trend; *Z*(t) the linear concentration of the main-shocks; and *B*(t) is the maximal number of aftershocks as a measure of EQ clustering (these functions are actually properly normalized). The algorithm then recognizes a well established criterion, defined by extreme values of the phase space coordinates, as a vicinity of the system singularity. When a trajectory enters the criterion, probability of extreme event increases to the level sufficient for its effective provision, so an alarm or a TIP, "Time of Increased Probability", is declared. This algorithm can be modifiedadapted for lower magnitudes and particular regions (e.g. CN8).

*The Reverse Tracing of Precursors (RTP)*

The RTP is a method for medium-term (some months in advance) EQ prediction (Shebalin et al., 2006), which is based on a hierarchical ensemble of premonitory seismicity patterns. These patterns are: (1) "precursory chains" that are related with the correlation length (e.g. Zoller and Hainzl, 2002, Tyupkin and Di Giovambattista, 2005), (2) "intermediate-term patterns" that could be related to some accelerating seismicity (e.g. Gabrielov et al., 2000) and (3) "pattern recognition of infrequent events" that take into account several "opinions" to decide the validity of the calculated chain of events. If a sufficient number of "votes" is accumulated, then the chain is considered precursory (Shebalin et al., 2006). Some past EQs seem to have been predicted 6 to 7 months in advance, although a few false alarms also happened. Critical aspects are related to the predicted "area of alarm" that seems very large for a realistic application. RTP has already evaluated by the gambling score, showing apparently only marginal or no significance in predicting earthquakes (Zechar and

Zhuang 2010). However, Molchan (2011) criticised this conclusion, affirming that; "The statistical analysis of any prediction method with few target events and a short monitoring period is premature (this is the case of RTP)".

[revised manuscript text omitted]

16. explanation of the typical decrease of b-value as seismic precursor (e.g. Sugan et al. 2014 for the

17. case of 6 April 2009 M6.3 L'Aquila earthquake in Central Italy) as an increase of entropy

18. culminating almost at the mainshock (De Santis et al. 2011a).

19.

20. **7. Entropy and critical point theory**

21.

22. An *ergodic* dissipative system can have a critical point where the system undergoes through a

23. transition. The ergodic property means that the system averages in real 3D space are equivalent to

24. averages in the ideal reconstructed phase space (e.g. Takens, 1981, De Santis et al., 2011b). As an

25. example, Fig. 1 reports the behaviour of the *specific heat* around a critical point occurring at

26. temperature $T_\lambda$ degree. It is interesting how the system approaches the critical temperature as a

27. power law. In addition, if the system changes its temperature linearly in time, the same plot is

28. expected versus time.

[Figure]

**Fig. 1**. Specific heat of $^4$He as a function of T-$T_\lambda$ in Kelvin. $T_\lambda$ is the temperature at which the critical system has a transition (critical point) (Adapted from Stanley, 1971).

More generally, if we replace the increasing temperature with the system entropy, then the system reaches its critical point (vertical red line in Fig.2) at the largest Entropy and approaches it with an accelerating power law in its cumulative of *punctuated* events (we intend here for an "event" as an anomalous behaviour of the system evolution, e.g. when its signal level is larger than a certain number of standard deviation, $\sigma$, e.g. 2.5 $\sigma$).

[Figure]

**Fig. 2**. Idealized Shannon entropy (above diagram) and cumulative number of events (bottom diagram) for a dissipative system around its critical point, indicated by the vertical red line.

After the critical point, the curve behaves as a decelerating power law. Fig. 2 depicts both the idealized behaviours for the entropy and the cumulative number of events.

We will see in the following how these patterns are reproduced in the case studies of some Italian seismic sequences.

**8. Entropy studies of two Italian seismic sequences**

In this part, we show two case studies in Italy: the 2009 L'Aquila and the 2012 Emilia seismic sequences, both producing a main-shock of around M6 (precisely local and moment magnitudes, ML5.9 and Mw6.2 for L'Aquila and local magnitude ML5.9 for Emilia). Main characteristics of the two seismic sequences are given in Table 1. The first case was already analysed and discussed by De Santis et al. (2011a). However, we will make here some alternative/complementary analyses, with respect to those already published. The second case study is original and never published so far. In both cases we considered all earthquakes with a miminum magnitude equal to (entropy analysis) or well above (R-AMR; see section 9) the completeness magnitude of the earthquake catalogs, that was found of M1.4+ and M2+ for L'Aquila and Emilia earthquake sequences, respectively.

[revised manuscript text omitted]

**Fig. 5**. Shannon entropy for Emilia seismic sequence from 2000 to 2014. The significant increase from around 2010,
with the maximum at around the main-shock occurrence, is expected to be real. The gray area defines the statistically
estimated (1 standard deviation) error in computing the entropy.
As a general remark of this section, it is true that we applied the entropy analysis to only two case
studies, but in most occasions, we could extend the results found analyzing $b$-value to the entropy,
via the equation (6). The introduction of the Shannon entropy in the analysis of a seismic sequence
provides a more physical and statistical meaning to the potential precursory decrease of the b-value
in terms of an increasing of entropy of the underlying physical system.

[revised manuscript text omitted]

The above cases represent two of the four seismic sequences happened in Italy in the last 15 years.

Another seismic sequence occurred in south Italy in 2010 and culminated with a M5 in Pollino area showing an analogous acceleration before the mainshock (De Santis et al., 2015). Only the most recent seismic sequence of the Amatrice-Norcia (Central Italy) earthquakes in 2017, did not have neither acceleration nor foreshocks before the first major earthquake (24 August 2016 M6 Amatrice earthquake). Therefore, the two case studies here shown are very representative of the most recent seismicity in Italy that has expressed in terms of a series of earthquakes culminating with a mainshock.

[revised manuscript text omitted]
. As example, Table 2 shows the confusion matrix of the validity of the skt as precursor applied to Central Italy earthquakes from 1994 to 2016. The following results are obtained: overall accuracy= 74%, hit rate of success= 40%, false alarms=17%, Cohen coefficient=0.23 (for more details on the corresponding definitions please see Piscini et al. 2017). These values confirm the validity of this thermal parameter as potential pre-earthquake indication, at least for the area of concern, i.e. Central Italy.

| skt | Seismicity | |
|---|---|---|
| | Yes | No |

| | | |
|---|---|---|
| Yes | 2 | 3 |
| No | 3 | 15 |

Table 2 Confusion matrix for pre/earthquake anomaly detection obtained from skt time series analysis since 1994 in Central Italy (adapted from Piscini et al., 2017).

**12. Mutual Information and Transfer Information: a possible future direction**

Geosystemics focuses on the inter-relations among the components composing the terrestrial complex system. For this reason, every statistical (or physical) quantity that measures these inter-relations is useful. However, given that the system under study is not usually linear, instead of linear quantities such as correlation coefficient or cross-correlation function between two variables belonging to linear processes, we have to resort to statistical quantities, which are more appropriate for nonlinear processes, as typical in a complex system.

Given two variables X and Y, characterising two processes of the phenomenon under study, we define the mutual information I(X, Y) extending definition (1) to two variables, i.e.:

$$I(\mathrm{X};Y) = \sum_{y \in Y} \sum_{x \in X} p\ (x,y) \cdot \log\left(\frac{p(x,y)}{p_1(x)p_2(y)}\right) \tag{10}$$

where $p_1(x)$ and $p_2(y)$ are the corresponding probabilities and p(x,y) is the joint probability.

However, this formulation does not provide hints about the direction of information transfer between process X and process Y, i.e. from a part of a system to another. For this purpose, it is possible to introduce a useful definition that quantifies the information flow in terms of the Kullback and Leibler entropy (Kullback and Leibler, 1951), which can be defined for a single process X as:

$$K_x = \sum_x p(x) \cdot \log[p(x)/q(x)] \tag{11}$$

The above quantity is the entropy related to the process X when a different probability $q(x)$ is used instead of the true $p(x)$, and then adapt it in order to be applied to two variables taking into account of a proper delay in one (Schreiber 2000). The so-called transfer information (changing its sign it becomes the transfer entropy) provides also the direction of the information flow.

Here, we do not describe more details but we just want to emphasise the importance of quantifying direction of information flow amongst different parts or processes of the system under study, because often it is more important to know where the flow of information is going, instead of just estimating the information that is exchanged by the whole process between internal components or external ones (Ahlswede et al. 2006).

Applying this concept to two different time series can be useful to say if one is the master quantity (that represents the causal process) and the other the affected one. In all cases, where we would like to compare/correlate a seismic sequence with possible atmospheric or ionospheric series of precursors, the calculation of the transfer information would provide a robust answer.

**13. Conclusions**

This paper has introduced the concepts of geosystemics, and then has shown its applications to some case studies. The spirit of geosystemics is to use some universal tools to look at some macroscopic quantities, such as the entropy, the Benioff strain, or the temperature to consequently deduce macroscopic properties of the physical system under study. An important frame is that of dynamical systems approaching a critical point, when the macroscopic properties of the system change dramatically. This could be the case of a sequence of EQs that culminates with a main-shock. Therefore, we showed some results obtained with the study of two recent Italian seismic sequences, the 2009 L'Aquila and the 2012 Emilia sequences.

It is obvious that, being the study of EQs a very complex problem, the more characterizing parameters are analysed, the more robust the result will be. A recent and extensive example of this approach is given by Wu et al. (2016) for the case of the 2009 L'Aquila seismic sequence.

A further question is how we can use the Big Data in geosciences and, in particular, to analyze precursory patterns of big earthquakes. Of course, the analysis of a greater number of data and the check of multiple models is perceptible that allows to find some type of pattern before an earthquake, that could be likely valid only for regions very localized. An extensive statistical big data analysis would be important to confirm or confute the individual case results (although no definitive conclusions can be arisen, because high correlation does not always mean causation; however can be of great help in proposing a physical framework of the chain of processes that could occur before a large earthquake). An example of this approach is given by Piscini et al. (2017) where the validity of local climatological variations as possible seismic precursors in Central Italy has been statistically established.

Finally, we hope that this investigation can contribute to the worldwide scientific debate and efforts in understanding the earthquake preparation phase in order to arm the scientific community and stakeholders  against the natural disasters.

[revised manuscript text omitted]

Tramutoli, V. , Using RST approach and EOS-MODIS radiances for monitoring seismically active regions: a study on the 6 April 2009 Abruzzo earthquake. *Nat. Hazards Earth Syst. Sci.* 10, 239–249, 2010.

Tyupkin, Y. S., Di Giovambattista, R. , Correlation length as an indicator of critical point behavior prior to a large earthquake, *Earth Planet. Sci. Lett.*, 230, 85 – 96, doi:10.1016/ j.epsl.2004.10.037, 2005.

Ulam S. M. and von Neumann J., On the Combinations of Stochastic and Deterministic Processes, *Bull. Amer. Math. Soc.* 53, 1120, 1947.

Utsu, T., Estimation of parameter values in the formula for the magnitude-frequency relation of earthquake occurrence, *Zisin* 31, 367-382, 1978.

Wang, L.-W., Shen, X.-H., Zhang, Y., Yan,R., Preliminary proposal of scientific data verification in CSES mission. *Earthquake Science* 28 (4): 303–310. doi: dx.doi.org/10.1007/s11589-015-0131-2, 2015.

Wu L.X., Zheng S., De Santis A., Qin K., Di Mauro R., Liu S.J. and Rainone M. L., Geosphere Coupling and Hydrothermal Anomalies before the 2009 Mw 6.3 2 L'Aquila Earthquake in Italy, *Nat. Hazards Earth Syst. Sci*., 16, 1859-1880, doi:10.5194/nhess-2015-346, 2016.

Xie, T. and Weiyu, M., Possible Thermal Brightness Temperature Anomalies Associated with the Lushan (China) MS7.0 Earthquake on 20 April 2013. *Earthquake Science* 28 (1): 37–47. doi:10.1007/s11589-014-0106-8, 2015.

Xiong, Pan, Xuhui Shen, Xingfa Gu, Qingyan Meng, Yaxin Bi, Liming Zhao, Yanhua Zhao, Yan Li, and Jianting Dong, Satellite Detection of IR Precursors Using Bi-Angular Advanced along-Track Scanning Radiometer Data: A Case Study of Yushu Earthquake. *Earthquake Science* 28 (1): 25–36. doi:10.1007/s11589-015-0111-6, 2015.

Xu, T., Hu, Y.L., Wang, F.F., Chen Z., Wu J., Is there any difference in local time variation in ionospheric F2 layer disturbances between earthquake induced and Q- disturbances events?, *Ann. Geophysicae*, 33, 687-695, 2015.

Zechar, J.D. & Zhuang, J., Risk and return: evaluating RTP earthquake predictions, Geophys. J. Int., 182, 1319–1326. 2010.

Zoller, G., Hainzl, S. A systematic spatiotemporal test of the critical point hypothesis for large earthquakes. *Geophys. Res. Lett.*, 29, 1558, 2002.